# 3D atomic-scale imaging of mixed Co-Fe spinel oxide nanoparticles during oxygen evolution reaction

Weikai Xiang[1], Nating Yang[2], Xiaopeng Li[3], Julia Linnemann[4], Ulrich Hagemann[5], Olaf Ruediger[6], Markus Heidelmann[5], Tobias Falk[7], Matteo Aramini[8], Serena DeBeer[6], Martin Muhler[7], Kristina Tschulik[4] & Tong Li[1✉]

The three-dimensional (3D) distribution of individual atoms on the surface of catalyst nanoparticles plays a vital role in their activity and stability. Optimising the performance of electrocatalysts requires atomic-scale information, but it is difficult to obtain. Here, we use atom probe tomography to elucidate the 3D structure of 10 nm sized $Co_2FeO_4$ and $CoFe_2O_4$ nanoparticles during oxygen evolution reaction (OER). We reveal nanoscale spinodal decomposition in pristine $Co_2FeO_4$. The interfaces of Co-rich and Fe-rich nanodomains of $Co_2FeO_4$ become trapping sites for hydroxyl groups, contributing to a higher OER activity compared to that of $CoFe_2O_4$. However, the activity of $Co_2FeO_4$ drops considerably due to concurrent irreversible transformation towards $Co^{IV}O_2$ and pronounced Fe dissolution. In contrast, there is negligible elemental redistribution for $CoFe_2O_4$ after OER, except for surface structural transformation towards $(Fe^{III}, Co^{III})_2O_3$. Overall, our study provides a unique 3D compositional distribution of mixed Co-Fe spinel oxides, which gives atomic-scale insights into active sites and the deactivation of electrocatalysts during OER.

[1] Institute for Materials, Ruhr-Universität Bochum, Universitätsstraße 150, 44801 Bochum, Germany. [2] CAS Key Laboratory of Low-Carbon Conversion Science and Engineering, Shanghai Advanced Research Institute (SARI), Chinese Academy of Sciences (CAS), 201210 Shanghai, China. [3] State Key Laboratory for Modification of Chemical Fibers and Polymer Materials and College of Materials Science and Engineering, Donghua University, 201620 Shanghai, China. [4] Faculty of Chemistry and Biochemistry, Analytical Chemistry II, Ruhr-Universität Bochum, Universitätsstraße 150, 44801 Bochum, Germany. [5] Interdisciplinary Center for Analytics on the Nanoscale (ICAN) and Center for Nanointegration Duisburg-Essen (CENIDE), University of Duisburg-Essen, Carl-Benz-Straße 199, 47057 Duisburg, Germany. [6] Max Planck Institute for Chemical Energy Conversion, Stiftstraße 34-36, 45470 Mülheim an der Ruhr, Germany. [7] Faculty of Chemistry and Biochemistry, Laboratory of Industrial Chemistry, Ruhr-Universität Bochum, Universitätsstraße 150, 44801 Bochum, Germany. [8] Diamond Light Source, Harwell Science and Innovation Campus, Chilton, Didcot OX11 0DE, UK. ✉email: tong.li@rub.de

Hydrogen has long been proposed as a clean energy carrier within sustainable energy infrastructure. Although water electrolysis is a key technology in the production of hydrogen, it remains inefficient, and there are many complex challenges to improve its efficiency. One of the major hurdles is the limitation in the performance of anode electrocatalysts, where the oxygen evolution reaction (OER) takes place[1,2]. Optimisation of OER electrocatalysts requires a detailed understanding of the correlation between the surface composition of electrocatalysts and their activity and stability. However, it is notoriously challenging to perform a full three-dimensional (3D) structural and chemical characterisation of the topmost atomic layers of electrocatalysts, especially for catalyst nanoparticles <100 nm in diameter. In addition, the electrocatalyst surfaces undergo drastic structural and compositional changes during OER. Therefore, to develop high-performance OER electrocatalysts, it is imperative to thoroughly evaluate the contribution made by individual atoms during reactions to the relationships between catalytic activity and stability.

Mixed 3d transition metal oxides, such as mixed Co-Fe spinel oxides, have attracted much attention in the context of OER electrocatalysts due to their high abundance, low cost and rich redox chemistry[3–5]; these characteristics make them attractive alternatives to the high-cost benchmark noble metal-based oxides, i.e., $IrO_2$ and $RuO_2$. Depending on the composition, two spinel structures can be formed: (i) spinel, whereby a divalent cation, e.g., $Co^{II}$, is located at the tetrahedral site, and trivalent $Fe^{III}$ at the octahedral site, and (ii) inverse spinel, whereby $Co^{II}$ is located at the octahedral site and $Fe^{III}$ at both the tetrahedral and octahedral sites[6]. The addition of small amounts of Fe in $Co_3O_4$ has been found to reduce the overpotential, while excess Fe increases the overpotential[7,8]. However, the role of Fe of mixed Co-Fe oxides or (oxy)hydroxides in catalysing OER is poorly understood, being the subject of ongoing and intense debate[4,5,9–14]. Additionally, although surface chemical and structural rearrangement of Co-based spinel oxides has been recently observed[4,15–20], the surface reconstruction or phase transformation responsible for the change in OER activity and stability has not yet been studied in-depth. Therefore, this study aims to (i) correlate changes in OER performance with structural and compositional evolution of $Co_2FeO_4$ spinel and $CoFe_2O_4$ inverse spinel, thereby elucidating their deactivation processes during OER, and (ii) pinpoint the role of Fe in the OER activity of mixed Co-Fe oxides.

In this work, we use atom probe tomography (APT), in conjunction with X-ray photoelectron spectroscopy (XPS), X-ray absorption spectroscopy (XAS), high-resolution transmission electron microscopy (HRTEM) and electrochemical impedance spectroscopy (EIS) to characterise the evolution of the oxidation state, structure and composition on the surfaces of $Co_2FeO_4$ and $CoFe_2O_4$ nanoparticles during cyclic voltammetry (CV) measurements under OER conditions. Comprehensive information regarding the surface state changes is obtained by the scale-bridging method, including oxidation state measurements of bulk volume and top surface layer (5–10 nm) of nanoparticles by XAS and XPS, respectively, along with nanoscale and atomic-scale elemental and structural characterisation of individual nanoparticles by APT and HRTEM. Our study reveals the presence of Co-rich and Fe-rich nanodomains, created by spinodal decomposition, in pristine $Co_2FeO_4$ and most likely in most mixed $Co_xFe_{(3-x)}O_4$ spinel oxides when $x$ is in the range of 1.1–2.7 due to the miscibility gap[21,22]. Interestingly, hydroxyl groups were trapped at the interface between the nanodomains, possibly yielding a significantly enhanced OER activity of pristine $Co_2FeO_4$ compared to $CoFe_2O_4$. During OER, different levels of Fe dissolution occur in the nanodomains of $Co_2FeO_4$, along with concurrent irreversible structural transformation towards $Co^{IV}O_2$, leading to a substantial decrease in the OER activity. In contrast, negligible Fe loss was observed for $CoFe_2O_4$. Instead, $(Fe^{III}, Co^{III})_2O_3$ was formed on the surface, further decreasing the OER activity of $CoFe_2O_4$. Overall, our 3D atomic-scale data, combined with X-ray- and electron-based microscopy and electrochemical data, show great promise for improving understanding of the complex structure-activity-stability relationships of electrocatalysts.

## Results

**Structure, size and morphology of spinel oxide nanoparticles.** $Co_2FeO_4$ and $CoFe_2O_4$ nanoparticles were synthesised by a hydrothermal method (see Methods). Both pristine nanoparticles have the standard cubic spinel structure ($Fd\bar{3}m$[23]), as confirmed by X-ray powder diffraction (XRD) (Supplementary Fig. 1). The size of pristine $Co_2FeO_4$ and $CoFe_2O_4$ nanoparticles is 10.3 ± 2.6 nm and 10.4 ± 2.7 nm, respectively, and both have a spherical shape (Supplementary Figs. 2a–d and 3a–d). The lattice constants of $Co_2FeO_4$ and $CoFe_2O_4$ nanoparticles, as measured from the selected area electron diffraction patterns shown in Supplementary Figs. 2e and 3e, is 8.60 Å and 8.69 Å, respectively. The difference in the lattice constants originates from the differences between Co/Fe contents and their radius ($Co^{3+}$ has a radius of 0.61 Å, which is slightly less than the $Fe^{3+}$ radius of 0.65 Å[24]); this is consistent with the XRD data (in Supplementary Fig. 1), whereby the diffraction peaks of $Co_2FeO_4$ are shifted to higher 2θ values compared to those of $CoFe_2O_4$.

**Electrochemical performance.** The electrocatalytic activity was measured by linear sweep voltammetry (LSV), using a scan rate of 10 mV/s on a rotating disk electrode (RDE), on $Co_2FeO_4$ and $CoFe_2O_4$ nanoparticles in the pristine state and after various CV cycles in 1.0 M KOH under OER conditions, Fig. 1a–d. Tafel slopes were derived from the LSV data, see Fig. 1e, f. The current density was normalised to surface areas determined by the Brunauer–Emmett–Teller (BET) method from $N_2$ physisorption measurements (Supplementary Fig. 4, additionally, the current density normalised to the geometric surface area of glassy carbon electrodes was provided in Supplementary Fig. 5). Ohmic drop ($iR_s$) correction ($R_s$ extracted from Nyquist plots) was applied to compensate for a lowering of the actual potential resulting at the electrode as compared to the nominally applied one due to current flux in the highly resistive system[5].

The LSV plots, shown in Fig. 1a, b, reveal that pristine $Co_2FeO_4$ exhibits a higher OER activity than pristine $CoFe_2O_4$, since the overpotential of $Co_2FeO_4$ (359 mV at 10 μA/cm²) is lower than that of $CoFe_2O_4$ (432 mV at 10 μA/cm²). Pristine $Co_2FeO_4$ has a Tafel slope of 43 ± 1 mV/dec (Fig. 1e), while pristine $CoFe_2O_4$ has a much larger Tafel slope of 79 ± 2 mV/dec (Fig. 1f), indicating that OER charge transfer kinetics are faster on pristine $Co_2FeO_4$ than on pristine $CoFe_2O_4$. The measured Tafel slope of pristine $Co_2FeO_4$ nanoparticles is also lower than most pristine $Co_3O_4$ and Co-based spinel oxide nanoparticles (~60 mV/dec)[19,25]. Increasing the number of CV cycles leads to a gradual deterioration in activity of both $Co_2FeO_4$ and $CoFe_2O_4$ (Fig. 1a, b, e, f). In particular, the Tafel slope of $Co_2FeO_4$ increases to 83 ± 2 mV/dec after 1000 cycles, which is almost double the Tafel slope of the pristine state, while the Tafel slope of $CoFe_2O_4$ increases slightly to 83 ± 1 mV/dec. Thus, despite the high OER activity of pristine $Co_2FeO_4$, its OER activity drops as the number of CV cycles increases, eventually reaching similar values as detected for the less active $CoFe_2O_4$.

Furthermore, $Co_2FeO_4$ exhibits pronounced redox couples during CV measurements (inset of Fig. 1c). Specifically, during

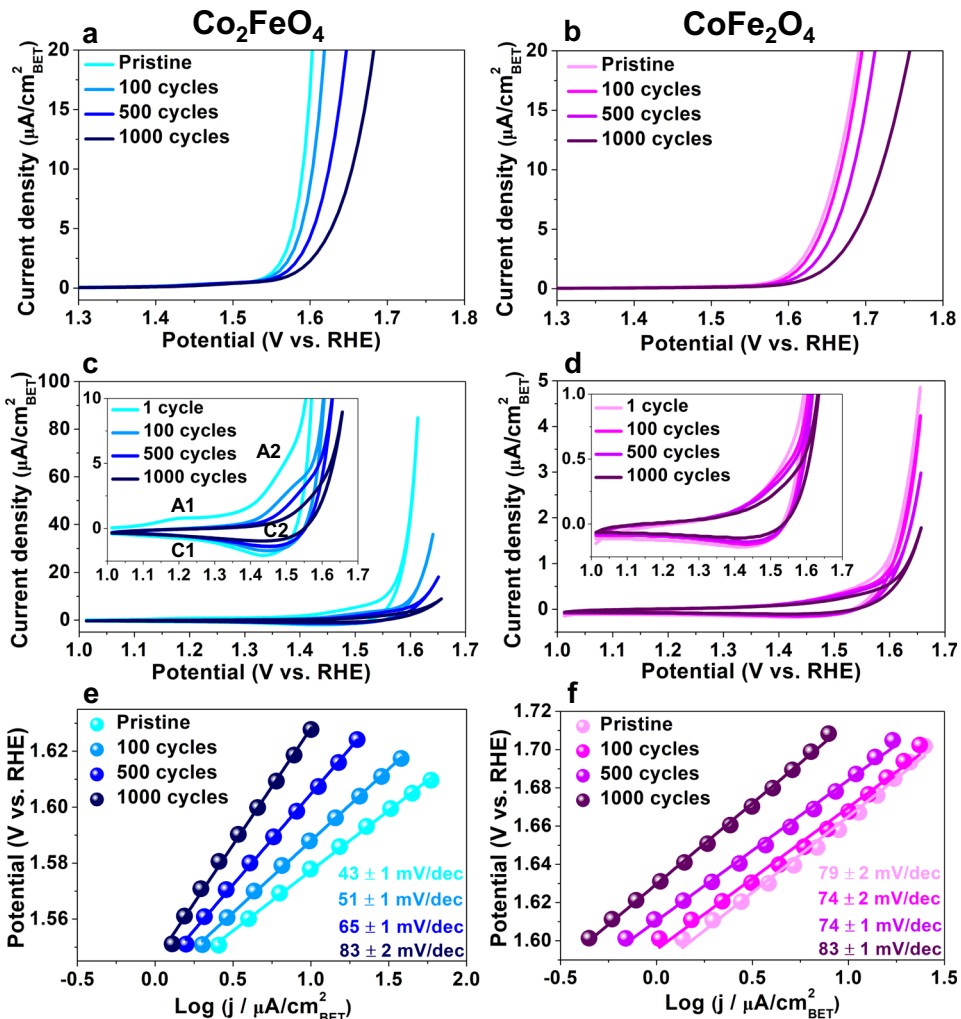

**Fig. 1 OER activity of Co₂FeO₄ and CoFe₂O₄ nanoparticles. a**, **b** Linear sweep voltammetry (LSV) curves recorded at a scan rate of 10 mV/s in 1.0 M KOH on glassy carbon electrodes deposited with $Co_2FeO_4$ and $CoFe_2O_4$ nanoparticles in the pristine state and after 100, 500 and 1000 cycles of cyclic voltammetry (CV) measurements, **c**, **d** CVs of $Co_2FeO_4$ and $CoFe_2O_4$ after one, 100, 500 and 1000 cycles recorded at a scan rate of 50 mV/s in 1.0 M KOH under OER conditions, **e**, **f** Tafel slopes of $Co_2FeO_4$ and $CoFe_2O_4$ in the pristine and after 100, 500 and 1000 cycles, derived from the LSV curves in **a**, **b**. Source data are provided as a Source Data file. The error bars of Tafel slopes in **e**, **f** were measured by linear curve fitting.

the first CV cycle, two broad anodic peaks are observed at ~1.19 V (A1) and ~1.48 V (A2), possibly corresponding to the oxidation process of Co(II)/Co(III) and Co(III)/Co(IV), respectively[26,27]. The cathodic sweep exhibits a relatively strong cathodic peak at ~1.44 V (C2), which is usually attributed to the Co(IV)/Co(III) couple in $Co_3O_4$[26–29]. The cathodic peak C1 at ~1.1 V for the Co(II)/Co(III) process[26,27] is almost negligible after the first CV cycle, suggesting that the Co(II)/Co(III) process is likely not to be fully reversible. Additionally, the A2 and C2 peaks become less pronounced and nearly indiscernible after 1000 cycles, which indicates that the Co(III)/Co(IV) oxidation is likely irreversible. The gradual formation of irreversible Co(III) and Co(IV) surface species possibly results in the A2 peak gradually shifting to higher potentials, which leads to the increased Tafel slope (Fig. 1e)[30]. In comparison with $Co_2FeO_4$, nearly no redox couples were observed for $CoFe_2O_4$, with a slight anodic shift after 1000 cycles (dark purple curve, insert in Fig. 1d), most likely suggesting the occurrence of an irreversible oxidation process.

**Oxidation state on the surfaces**. To investigate the reasons for the activity changes of $Co_2FeO_4$ and $CoFe_2O_4$, we first performed XPS to examine the oxidation state of Co and Fe on the surface of

$Co_2FeO_4$ and $CoFe_2O_4$ in their pristine state as well as after 100, 500, and 1000 cycles. XPS measures the average oxidation state of approx. 100 μm × 100 μm × 5 nm of the surface region of the nanoparticles deposited on glassy carbon. Given the closeness of $2p_{1/2}$ and $2p_{3/2}$ peak locations for Co(II), Co(III) and Co(IV)[31–34], the satellite features and their intensity change during OER, i.e., 786.5 eV for CoO-like Co(II)[35] and 789.5 eV for $Co_3O_4$-like Co (II, III), were analysed (peak fitting shown in Supplementary Fig. 6a). We can see from Fig. 2a that the intensity of CoO-like Co(II) satellite features decreases after 100 cycles, suggesting the oxidation of Co(II) to Co(III). As the number of CV cycles increases, the contribution of CoO-like Co(II) decreases significantly, and $Co_3O_4$-like Co increases (as indicated by the depth analysis using peak deconvolution of the Co 2p peak[35] shown in Supplementary Fig. 6b). Our CV data for $Co_2FeO_4$, shown in Fig. 1c, indicate an irreversible oxidation of Co(II)/Co(III) in the first cycle and of Co(III)/Co(IV) after 1000 cycles. While the presence of Co(IV) is difficult to be confirmed by XPS, the possibility cannot be excluded since the spectrum of $Co_3O_4$-like Co is similar to that of $CoO_2$-like Co(IV)[36]. Thus, we conclude that Co(III) and Co(II) are present on the surface of $Co_2FeO_4$ after 100 and 500 cycles, while Co(III) and Co(IV) are likely present

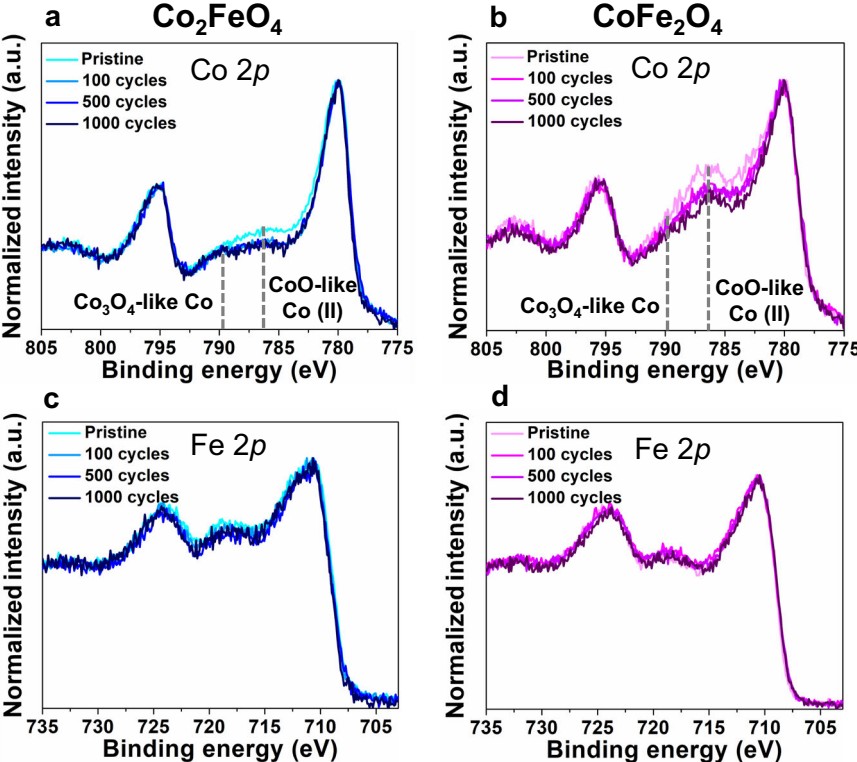

**Fig. 2 Surface oxidation state of Co₂FeO₄ and CoFe₂O₄ during OER.** X-ray photoelectron spectra of **a**, **b** Co 2*p* and **c**, **d** Fe 2*p* levels of Co₂FeO₄ and CoFe₂O₄ in the pristine state and after 100, 500 and 1000 cycles. The dashed lines indicate the satellite features of Co₃O₄-like Co (II, III) and CoO-like Co(II). Source data are provided as a Source Data file.

after 1000 cycles. In contrast to Co₂FeO₄, the CoO-like Co(II) satellite peaks of CoFe₂O₄, in Fig. 2b, decreases less significantly as the CV cycles increases. Co(III) seems to be present on the surface of CoFe₂O₄ after 500 and 1000 cycles, as shown in Supplementary Fig. 6c. In addition to Co, the Fe 2*p* spectra, shown in Fig. 2b, d remains unchanged, indicating that Fe(III) is present on the surface of both Co₂FeO₄ and CoFe₂O₄ as the number of CV cycles increases, in agreement with observations in most previous studies[4,13].

To further verify the irreversible change in the oxidation state of Co₂FeO₄ in their pristine state and after 1000 cycles, we performed XAS that allows spectral detection of a bulk volume of approx. 100 μm × 300 μm × 1 mm (penetration depth) of nanoparticles deposited on glassy carbon (Supplementary Note 1 and Supplementary Fig. 7). We observed a subtle shift of Co K-edge towards higher energy values (Supplementary Fig. 7a), possibly suggesting that only a small volume fraction, potentially on the surface regions, has an increase in oxidation state to Co(IV), which is in agreement with the XAS data of Calvillo et al.[20]. Additionally, more octahedrally coordinated and less tetrahedrally coordinated Co[37] was observed after 1000 cycles (inset, Supplementary Fig. 7a). By relating XPS and electrochemical data (Figs. 1c and 2a), we, therefore, speculate that tetrahedrally coordinated Co(II) irreversibly oxidises to octahedrally coordinated Co(III) or (IV) in the course of 1000 cycles, yielding a decrease in activity.

**Structural changes.** Next, HRTEM was employed to evaluate the structural changes of the Co₂FeO₄ and CoFe₂O₄ nanoparticles after OER, first in their pristine state and then after 100 and 1000 cycles. Figure 3a shows a background-subtracted HRTEM image of pristine Co₂FeO₄, viewed along the [001] zone axis. Interestingly, the interplanar spacing of d₂₂₀ varies slightly in different

regions, as presented in Fig. 3a (details in Supplementary Fig. 8). After 100 cycles, a distinct structural change occurs on the surface of Co₂FeO₄, as highlighted by the blue-dotted areas in Fig. 3b (more HRTEM images shown in Supplementary Fig. 9a, b). The Fast Fourier filtered transform (FFT) images, shown in insets of Fig. 3b, indicate that the motifs observed on the surface (blue-dotted region) are aligned at 45° to the atomistic arrangement of the spinel structure. The reflection spots from the surface region (Fig. 3b, bottom inset) correspond well with β-CoOOH (R̄3m, hexagonal[38], Supplementary Table 1), which agrees with observations of a previous study[19]. Therefore, we hypothesise that epitaxial growth of (Co, Fe)OOH occurs on Co₂FeO₄, with an orientation relationship of (010) Co₂FeO₄//(1-101) (Co, Fe)OOH. A previous study proposed that Co[II] ions at the tetrahedral site are oxidised to form amorphous Co[III] oxyhydroxides[16]. Here, we observed crystalline Co[III] oxyhydroxides, possibly formed by crystallisation of the amorphous Co[III] oxyhydroxides in the absence of potentials and electrolytes (also under high vacuum in TEM). Furthermore, after 1000 cycles, a 5–6 nm surface region, highlighted by the dark-blue-dotted area in Fig. 3c (Supplementary Fig. 9c, d), undergoes a phase transformation, since it contains a distinct lattice fringe from the [112]-oriented Co₂FeO₄ spinel oxide. The reflection spots in the FFT image (Fig. 3c, bottom inset) match well with the CoO₂ phase (P̄3m1, hexagonal[39], Supplementary Table 1), with an octahedrally coordinated Co (IV). The formation of CoO₂ is also confirmed by additional reflection spots that correspond well to (11–20) CoO₂ in the selected area electron diffraction (SAED) pattern after 1000 cycles (Supplementary Fig. 2f); the SAED pattern was recorded from a 500 nm × 500 nm area containing more than 100 nanoparticles. We observed an increase in octahedrally coordinated Co by XANES (Supplementary Fig. 7a, inset), the presence of irreversible Co(IV) by electrochemical data (Fig. 1a) and possible

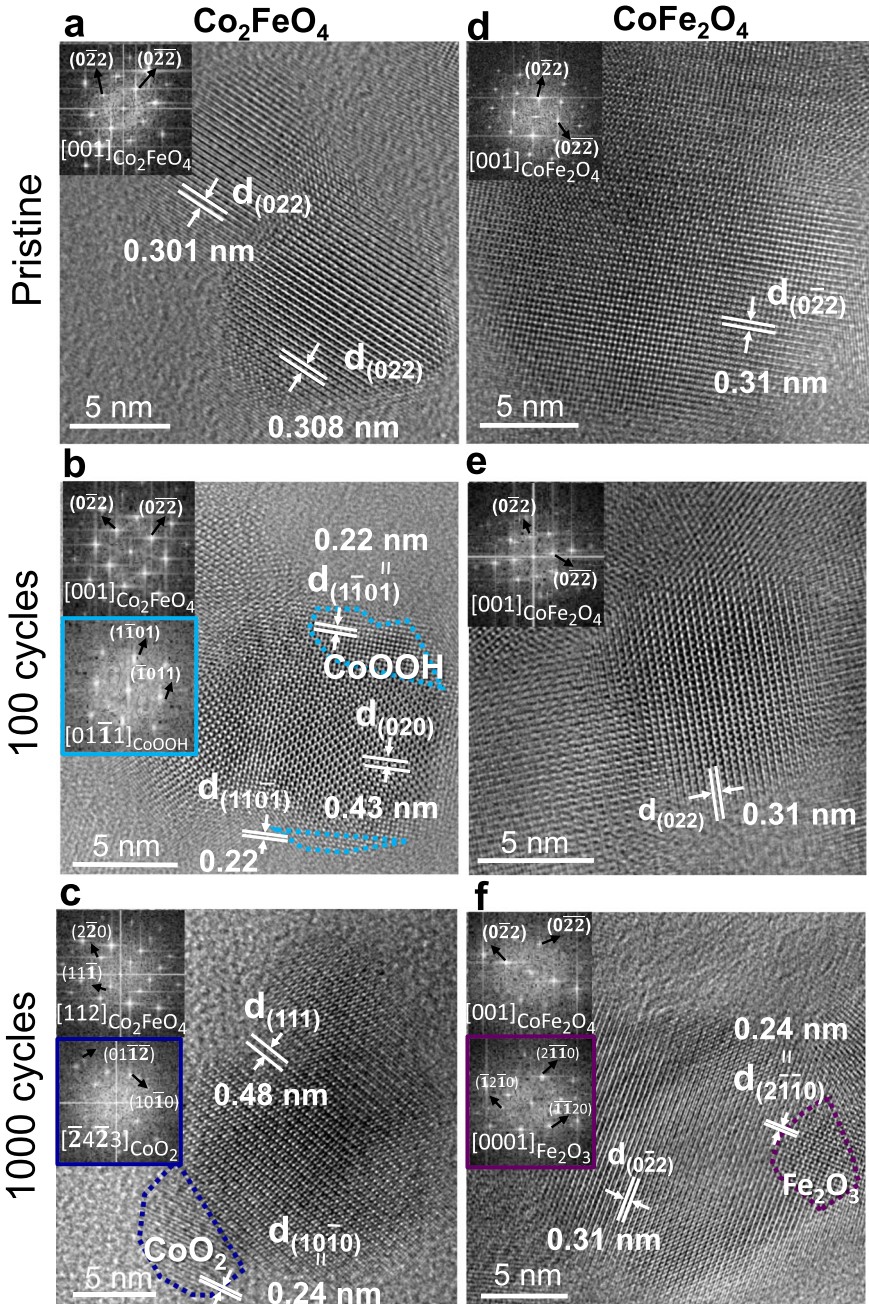

**Fig. 3 Surface structural evolution of Co₂FeO₄ and CoFe₂O₄ during OER.** High-resolution TEM images of Co₂FeO₄ and CoFe₂O₄ **a**, **d** in the pristine state, **b**, **e** after 100 cycles and **c**, **f** after 1000 cycles. The top insets in **a**–**f** were Fourier filtered transform images obtained from the centre of nanoparticles, and the bottom insets in **b**, **c**, **f** were recorded from the surface regions of nanoparticles highlighted by the dashed lines, indicating the occurrence of phase transformation on the surface regions of Co₂FeO₄ and CoFe₂O₄ during OER.

presence of Co (IV) by XPS (Fig. 2a). These results most likely imply an irreversible transformation towards (Co, Fe)O₂ phase on the surface of Co₂FeO₄ after 1000 cycles. Additionally, CoO₂ is the stable phase at higher potentials, in accordance with the Co Pourbaix diagram[40].

In contrast with Co₂FeO₄, we observed no significant structural changes on the CoFe₂O₄ nanoparticle surface after 100 cycles (Fig. 3d, e). After 1000 cycles, a structural transformation of the surface of [001]-oriented CoFe₂O₄ spinel nanoparticles was discerned, as indicated by the purple dotted lines in Fig. 3f (Supplementary Fig. 9e, f). The reflection spots in the bottom inset of Fig. 3f correspond to either [412]-oriented FeOOH (I4/m, tetragonal[41], Supplementary Table 1) or [0001]-orientated Fe₂O₃

phase (R3̄c, hexagonal[42]), with the latter having a better fit. Also, the SAED pattern of CoFe₂O₄, in Supplementary Fig. 3f, reveals additional reflection spots after 1000 cycles, indicating the presence of (30-30) Fe₂O₃ or (202) FeOOH. According to the Fe Pourbaix diagram[43], Fe₂O₃ forms as the potential increases. Thus, the surface of inverse spinel CoFe₂O₄ is thought to undergo a phase transformation to the (Fe, Co)₂O₃ phase after 1000 cycles.

Notably, we observed an amorphous layer on the surfaces of aggregated Co₂FeO₄ and CoFe₂O₄ nanoparticles after 100, 500 and 1000 cycles (exemplified in Supplementary Fig. 10). However, these amorphous layers were most likely the result of carbon contamination under electron beam in TEM (details in Supplementary Fig. 10).

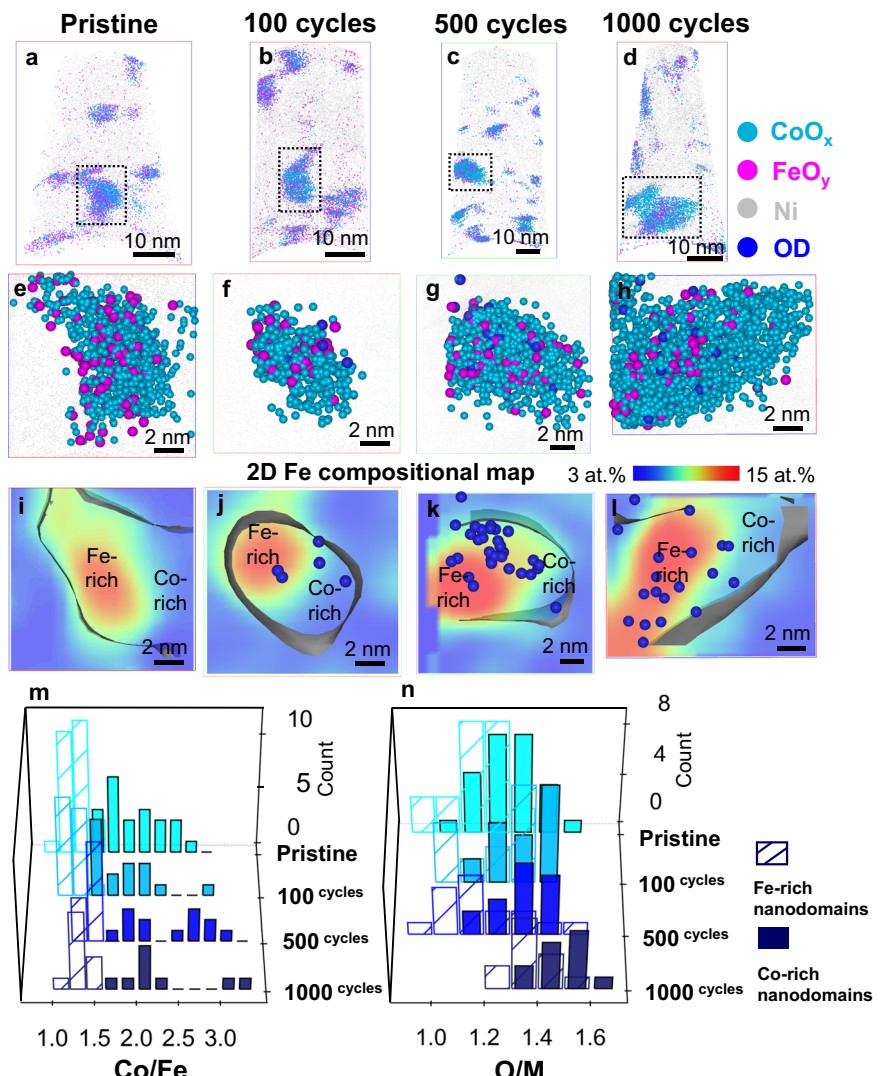

**Fig. 4 Compositional evolution and elemental distribution of segregated Co$_2$FeO$_4$ during OER. a–d** 3D-APT reconstruction of Co$_2$FeO$_4$ in the pristine state and after 100, 500 and 1000 cycles of CV showing the nanoparticles embedded in Ni matrix, **e–h** 3D atom maps of segregated Co$_2$FeO$_4$ nanoparticles in the pristine state and after 100, 500 and 1000 cycles, selected from dashed line region in a–d (**f** displayed in the top-down view of **b**), and **i–l** 2D Fe compositional maps of the same nanoparticles in **e-h** showing the Co-rich and Fe-rich nanodomains, compositional histograms of **m** Co/Fe ratios and **n** O/M ratios (M = Co+Fe) in the Fe-rich and Co-rich regions of segregated Co$_2$FeO$_4$ nanoparticles in **a–d** and Supplementary Fig. 12a–d. (Source data of Fig. 4m, n are provided as a Source Data file).

**Compositional evolution and correlation with electrochemical performance**. Although the (Co, Fe)OOH, which is observed on the surface of Co$_2$FeO$_4$ (Fig. 3b), can be regarded as an active intermediate for OER[27], the activity of Co$_2$FeO$_4$ decreased after 100 cycles (Fig. 1a, e). The cause of the decrease in the OER activity of Co$_2$FeO$_4$ (Fig. 1a, e) at the beginning of OER remains unclear. Other factors, such as chemical composition change, potentially lead to decreases in OER activity. Therefore, we used APT[44], a mass-spectrometry technique with sub-nanometre spatial resolution in three dimensions[45,46], to investigate the compositional evolution of oxide nanoparticles after OER. All electrochemical measurements were carried out in proton-free, deuterated electrolyte (i.e., 1.0 M KOD in D$_2$O in order to use APT to examine the distribution of hydroxyl groups after OER[47].

Figure 4a exemplifies a cross-sectional atom map of pristine Co$_2$FeO$_4$ nanoparticles embedded in a Ni matrix (APT specimen preparation is detailed in Supplementary Note 2/Supplementary Fig. 11 and additional APT data is shown in Supplementary

Fig. 12a). The oxide nanoparticles were detected by APT in the form of O ions and Co- and Fe- containing complex molecular ions (see mass spectra in Supplementary Fig. 13). All Co- (in blue) and Fe- (in magenta) containing molecular ions were shown as CoO$_x$ and FeO$_y$ respectively, in Fig. 4a (separate Co, Fe, O and Ni atom maps are shown in Supplementary Fig. 14). Our detailed APT analysis of 48 pristine Co$_2$FeO$_4$ nanoparticles (Fig. 4a and Supplementary Fig. 12a) reveals that 26 of them have nanoscale compositional modulation, while the remaining 22 exhibits a relatively uniform elemental distribution, as exemplified by Figs. 4e and 5a, selected from the black and red dashed boxes in Fig. 4a and Supplementary Fig. 12a, respectively; we term these nanoparticles 'segregated' and 'non-segregated' Co$_2$FeO$_4$ nanoparticles, respectively. The 2D Fe composition map (Fig. 4i), plotted from the nanoparticle data of Fig. 4e, clearly reveals separate Fe-rich and Co-rich nanodomains, whose dimensions are in the range of 4–5 nm, in the segregated pristine Co$_2$FeO$_4$ nanoparticle. In contrast, non-segregated pristine Co$_2$FeO$_4$ nanoparticles have uniformly distributed Fe and Co (Fig. 5a,e).

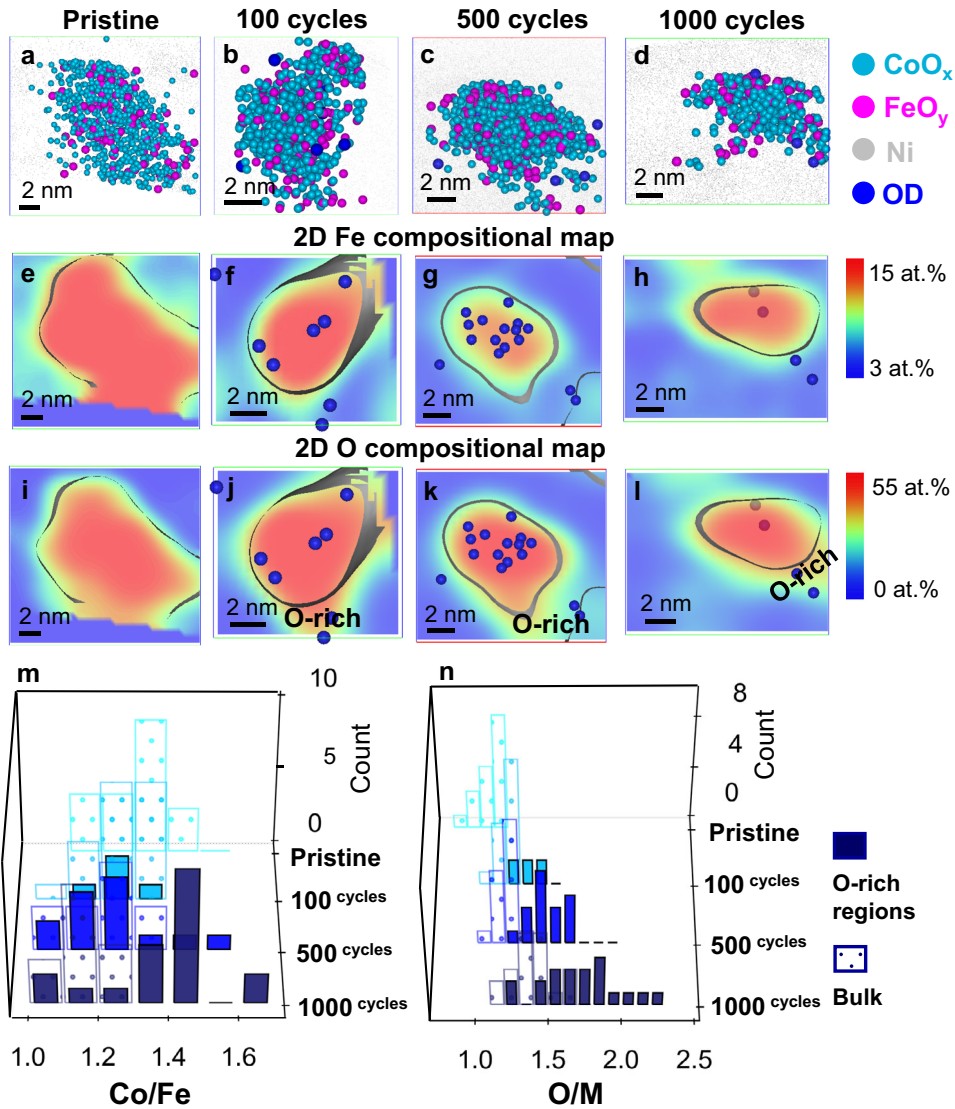

**Fig. 5 Compositional evolution and elemental distribution of non-segregated Co$_2$FeO$_4$ during OER. a–d** 3D atom maps, 2D compositional maps of **e–h** Fe and **i–l** O of non-segregated Co$_2$FeO$_4$ nanoparticles in the pristine state and after 100, 500 and 1000 cycles, compositional histograms of **m** Co/Fe ratios and **n** O/M ratios (M = Co+Fe) in the O-rich nanoregions and the bulk part of non-segregated Co$_2$FeO$_4$ nanoparticles in Fig. 4a–d and Supplementary Fig. 12a–d. (Source data of Fig. 5m, n are provided as a Source Data file).

The compositions of the Fe-rich and Co-rich nanodomains in pristine Co$_2$FeO$_4$, obtained by 1D composition profiles in Supplementary Figs. 15a and 16a and data from 25 segregated nanoparticles in Supplementary Fig. 12a, were plotted as composition histograms of Co/Fe and oxygen/(Co+Fe) (termed O/M) ratios in Fig. 4m, n. The average ratios of Co/Fe and O/M from all nanoparticles are detailed in Tables 1 and 2. Note that the oxygen content was normalised by the value measured by using H$_2$ temperature-programmed reduction (H$_2$ TPR) and the effect of laser pulse energy on the measurement of oxide stoichiometry[48] was detailed in Supplementary Note 3 and Supplementary Figs. 22 and 23. The Fe-rich and Co-rich nanodomains have Co/Fe ratios of $1.2 \pm 0.1$ and $2.2 \pm 0.1$, respectively, and O/M ratios of $1.2 \pm 0.1$ and $1.3 \pm 0.1$, respectively. Thus, although the Fe-rich and Co-rich nanodomains have similar oxygen contents, they have significantly different Co/Fe ratios; the stoichiometry of the 'segregated' Co$_2$FeO$_4$ nanoparticle is Co$_{2.1}$Fe$_{0.9}$O$_{3.9}$ in the Co-rich nanodomain and Co$_{1.65}$Fe$_{1.35}$O$_{3.6}$ for the Fe-rich nanodomain. Similarly, the stoichiometry of the non-segregated nanoparticles is Co$_{1.7}$Fe$_{1.3}$O$_{3.3}$ (based on values

given in Tables 1 and 2). The formation of Fe-rich and Co-rich nanodomains in the pristine Co$_2$FeO$_4$ nanoparticles is most likely the result of spinodal decomposition that is driven by the miscibility gap in the composition range $0.37 < \text{Co}/(\text{Co}+\text{Fe}) < 0.9$ at temperatures below 700 °C[21,22]. According to the CoFe$_2$O$_4$-Co$_3$O$_4$ phase diagram[21,22], Co$_2$FeO$_4$ is expected to decompose to Co$_{1.4}$Fe$_{1.6}$O$_4$ (Co/Fe = 0.875) and Co$_{2.4}$Fe$_{0.6}$O$_4$ (Co/Fe = 4). The discrepancy of compositions between our study and previous work possibly arises from the fact that these nanoparticles do not reach thermodynamic equilibrium after synthesis, and we expect the phase stability of nanoparticles to deviate from that of bulk materials in previous studies[21,22]. Despite this, we unambiguously reveal nanoscale compositional modulation of Co$_2$FeO$_4$ nanoparticles. In contrast, we did not observe any segregation for the CoFe$_2$O$_4$ nanoparticles, as shown in the atom map of Fig. 6a and the 2D Fe compositional map of Fig. 6e, since the Co/(Co+Fe) ratio (0.33) falls outside the composition window of spinodal decomposition[21,22].

Next, we examined elemental redistribution of 'segregated' Co$_2$FeO$_4$ nanoparticles after 100, 500 and 1000 cycles under OER

**Table 1 Average Co/Fe ratios in $Co_2FeO_4$ and $CoFe_2O_4$ nanoparticles during OER calculated by the total number of Co and Fe counts in all analysed datasets.**

| | $Co_2FeO_4$ | | | | $CoFe_2O_4$ | |
| | Segregated | | Non-segregated | | | |
| Co/Fe | Fe-rich Nanodomains | Co-rich nanodomains | Bulk | O-rich regions | Bulk | O-rich regions |
|---|---|---|---|---|---|---|
| Pristine | 1.2 ± 0.1 | 2.2 ± 0.1 | 1.3 ± 0.1 | — | 0.49 ± 0.01 | — |
| 100 cycles | 1.3 ± 0.1 | 2.2 ± 0.1 | 1.3 ± 0.1 | 1.3 ± 0.1 | 0.48 ± 0.01 | — |
| 500 cycles | 1.5 ± 0.1 | 2.8 ± 0.1 | 1.3 ± 0.1 | 1.3 ± 0.1 | 0.48 ± 0.01 | 0.47 ± 0.01 |
| 1000 cycles | 1.4 ± 0.1 | 2.9 ± 0.1 | 1.2 ± 0.1 | 1.4 ± 0.1 | 0.48 ± 0.01 | 0.52 ± 0.01 |

The error bars for the ratio were calculated from $R\sqrt{(\frac{\sigma_a}{a})^2 + (\frac{\sigma_b}{b})^2}$ where $R$ is the ratio of Co/Fe, $a$ and $b$ is Co and Fe concentration and $\sigma_a$ and $\sigma_b$ is the standard deviation of Co and Fe concentration. (Source data is provided in a Source Data file).

**Table 2 Average O/M ratios in $Co_2FeO_4$ and $CoFe_2O_4$ nanoparticles during OER calculated by the total number of O and (Co+Fe) counts in all analysed datasets.**

| | $Co_2FeO_4$ | | | | $CoFe_2O_4$ | |
| | Segregated | | Non-segregated | | | |
| O/M | Fe-rich nanodomains | Co-rich nanodomains | Bulk | O-rich regions | Bulk | O-rich regions |
|---|---|---|---|---|---|---|
| Pristine | 1.2 ± 0.1 | 1.3 ± 0.1 | 1.1 ± 0.1 | — | 1.2 ± 0.1 | — |
| 100 cycles | 1.3 ± 0.1 | 1.4 ± 0.1 | 1.2 ± 0.1 | 1.4 ± 0.1 | 1.2 ± 0.1 | — |
| 500 cycles | 1.3 ± 0.1 | 1.4 ± 0.1 | 1.2 ± 0.1 | 1.5 ± 0.1 | 1.3 ± 0.1 | 1.7 ± 0.1 |
| 1000 cycles | 1.5 ± 0.1 | 1.6 ± 0.1 | 1.3 ± 0.1 | 1.8 ± 0.1 | 1.3 ± 0.1 | 1.8 ± 0.1 |

The error bars for the ratio were calculated from $R\sqrt{(\frac{\sigma_a}{a})^2 + (\frac{\sigma_b}{b})^2}$ where $R$ is the ratio of O/(Co+Fe), $a$ and $b$ are O and (Co+Fe) concentration and $\sigma_a$ and $\sigma_b$ are their standard deviation. (Source data is provided in a Source Data file).

conditions (Fig. 4b–d, f–h). Importantly, we observed that the hydroxyl groups, shown as dark blue spheres in 2D Fe compositional profiles of Fig. 4j–l, are preferentially located at the interface between Fe-rich and Co-rich nanodomains after 100 and 500 cycles, and dominantly in Fe-rich regions after 1000 cycles. The trapping of hydroxyl groups at the interface of two nanodomains is most likely induced by the elastic strain that results from the difference in lattice constants of these two domains. This is confirmed by the difference in interplanar spacing of $d_{220}$ measured from two regions of pristine $Co_2FeO_4$ nanoparticles (Fig. 3a and Supplementary Fig. 8), which is ascribed to the difference in Co/Fe content of these nanodomains (similar to the difference of lattice constants of pristine $Co_2FeO_4$ and $CoFe_2O_4$ observed by XRD shown in Supplementary Fig. 1). The hydroxyl groups can be considered as 'fingerprints' that indicate the regions where OER occurs[49]. Therefore, we hypothesise that interfaces between two nanodomains provide active sites and accelerate OER kinetics, thereby possibly contributing to the high OER activity of pristine $Co_2FeO_4$ nanoparticles.

More importantly, we observed a dramatic compositional change within the nanodomains of $Co_2FeO_4$ as the number of CV cycles increased (Supplementary Figs. 15b–d and 16b–d along with all other nanoparticle data in Supplementary Fig. 12b–d were summarised in Fig. 4m, n and Tables 1 and 2; the number of nanoparticles for composition histograms was detailed in Supplementary Table 2). Specifically, the Co/Fe ratio in the Co-rich nanodomains remained at ~2.2 after 100 cycles but increased to ~2.8 after 500 cycles and 2.9 after 1000 cycles (Table 1). This result suggests a gradual Fe loss in the Co-rich nanodomains during OER (as also confirmed by 1D profiles of atomic counts in Supplementary Fig. 15e–h). The O/M ratio in Co-rich nanodomains increases gradually to ~1.4 after 100 and 500 cycles, and ~1.6 after 1000 cycles (Table 2 and Fig. 4n). The O/M ratio in the

Fe-rich nanodomains also increases to ~1.5 after 1000 cycles. The gradual increase in the O/M ratio of the segregated $Co_2FeO_4$ nanoparticles in both Co-rich and Fe-rich nanodomains suggests the occurrence of oxidation in both nanodomains. A more pronounced Fe loss was observed in the Co-rich nanodomains compared to that of the Fe-rich nanodomains (Table 1 and Fig. 4m), possibly suggesting that OER occurs more rigorously in the Co-rich nanodomains than that in Fe-rich nanodomains.

For the 'non-segregated' $Co_{1.7}Fe_{1.3}O_{3.3}$ nanoparticles, we observed 2–3 nm oxygen-rich surface regions after 100 cycles by comparing the 2D Fe and O compositional maps (Fig. 5f, j) of the same nanoparticle. The O/M ratio in the oxygen-rich regions increases from 1.4 ± 0.2 after 100 cycles to 1.8 ± 0.2 after 1000 cycles (see Table 2 and Fig. 5n, which was measured from the 1D concentration profiles in Supplementary Figs. 17 and 18, and all other nanoparticle data in Supplementary Fig. 12b–d). Based on the O/M ratios listed in Table 2, we speculate that the oxygen-rich surface regions possibly correspond to (Co, Fe)OOH after 100 cycles, and (Co, Fe)O2 after 1000 cycles, as observed by HRTEM (Fig. 3b, c and Supplementary Fig. 9a–d). Additionally, we observed a subtle Fe loss in the oxygen-rich region after 1000 cycles, as the Co/Fe ratio increases (Table 1 and Fig. 5m). Previous work also observed an increasing Co/Fe ratio on the surface of $CoFe_{0.75}Al_{1.5}O_4$ by electron energy loss spectroscopy[5], attributing it to the formation of Co oxyhydroxide. Here, we speculate that the increasing Co/Fe ratio is most likely due to Fe loss during structural transformation under the OER conditions. Concurrent structural transformation and Fe dissolution most likely lead to the overall reduction in OER activity of $Co_2FeO_4$ after 100 and 500 cycles (Fig. 1a, c). After 1000 cycles, surface formation of stable $(Co^{IV}, Fe^{III})O_2$ further decreases the activity of $Co_2FeO_4$.

For comparison with $Co_2FeO_4$, the compositional changes of the $CoFe_2O_4$ nanoparticles after 100, 500 and 1000 cycles are

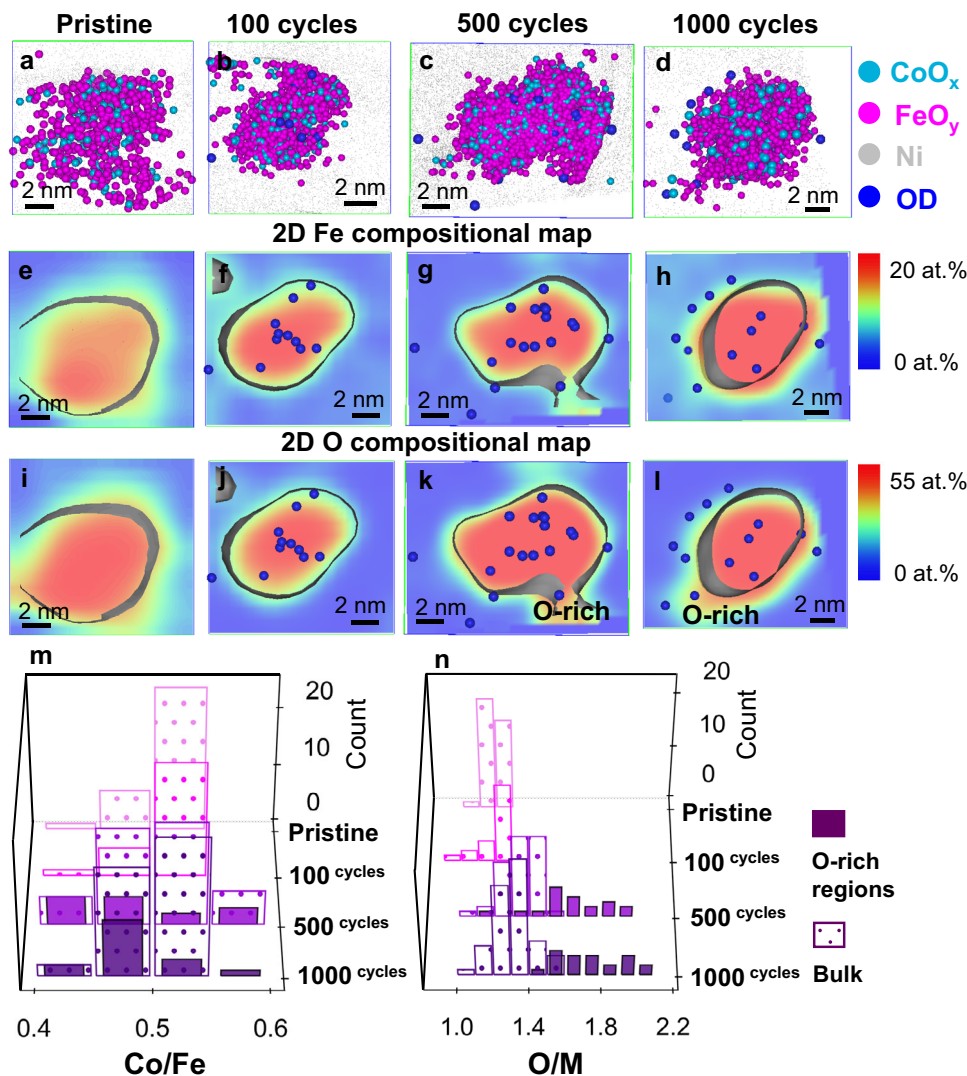

**Fig. 6 Compositional evolution and elemental distribution of CoFe$_2$O$_4$ during OER. a–d** 3D atom maps, 2D compositional maps of **e–h** Fe and **i–l** O of non-segregated CoFe$_2$O$_4$ nanoparticles in the pristine state and after 100, 500 and 1000 cycles, combined compositional histograms of **m** Co/Fe ratios and **n** O/M ratios (M = Co+Fe) in the O-rich nanoregions and the bulk part of non-segregated CoFe$_2$O$_4$ nanoparticles in Supplementary Fig. 19a–d. (Source data of Fig. 6m, n are provided as a Source Data file).

provided in Fig. 6b–l. The oxide stoichiometry of pristine CoFe$_2$O$_4$ is measured as CoFe$_2$O$_{3.6}$, based on values in Tables 1 and 2 (measured from 39 nanoparticles in Supplementary Fig. 19a and Supplementary Table 2). No evident elemental redistribution was observed after 100 cycles, while 2–3 nm oxygen-rich regions were seen on the surface of CoFe$_2$O$_4$ after 500 and 1000 cycles, as indicated by the 2D Fe and O compositional maps in Fig. 6g, h, k, l. The O/M ratio in the oxygen-rich regions increased to ~1.7 after 500 cycles, reaching up to ~1.8 after 1000 cycles, while Co/Fe increased only slightly (Tables 1 and 2 and Fig. 6m, n, derived from 1D concentration profiles in Supplementary Figs. 20 and 21 and all other data in Supplementary Fig. 19b–d). The XPS and XANES data suggest that Co$^{II}$ was oxidised to Co$^{III}$ while Fe$^{III}$ remained stable even after 1000 cycles (Fig. 2 and Supplementary Fig. 7). Therefore, we speculate that the oxygen-rich regions possibly correspond to the (Fe$^{III}$, Co$^{III}$)$_2$O$_3$ phase, as also observed by HRTEM (Fig. 3f and Supplementary Fig. 9e, f).

**Electrochemical sub-processes during OER.** To further understand the deactivation of both spinel oxide nanoparticles, we employed EIS to reveal the electrochemical sub-processes during OER. The Nyquist plots, in Fig. 7a, b, show two distorted semicircles for Co$_2$FeO$_4$, whereas one semicircle is observed for the inverse spinel CoFe$_2$O$_4$. The distribution of relaxation times for Co$_2$FeO$_4$ (inset of Fig. 7a) reveals three distinct peaks, and one peak is discerned for CoFe$_2$O$_4$ (inset of Fig. 7b). Figure 7c, d contain the resistances and capacitances for both spinel oxides in the pristine state and after 100, 500 and 1000 cycles (derived by equivalent circuit fitting[50,51], more details in Supplementary Note 4 and Supplementary Figs. 24 and 25). The equivalent circuit for Co$_2$FeO$_4$, shown in Fig. 7c, contains the double layer capacitor $C_1$ and the pseudocapacitors $C_2$ and $C_3$. The corresponding pseudocapacitive properties are based on changing oxidation states of electrochemically accessible cobalt sites and adsorptive discharge of oxygen-containing species in the electrolyte[52,53]. So, in course of a catalytic cycle, intermediate catalytic (re-)transformations occur via pseudocapacitive charging/discharging through the faradaic resistors $R_1$ and $R_2$[54]. In parallel, the OER reaction proceeds via OH$^-$-to-O$_2$ conversion steps at the solution side[55]. We can see from Fig. 7c that the resistance increases with the number of CV cycles, particularly for

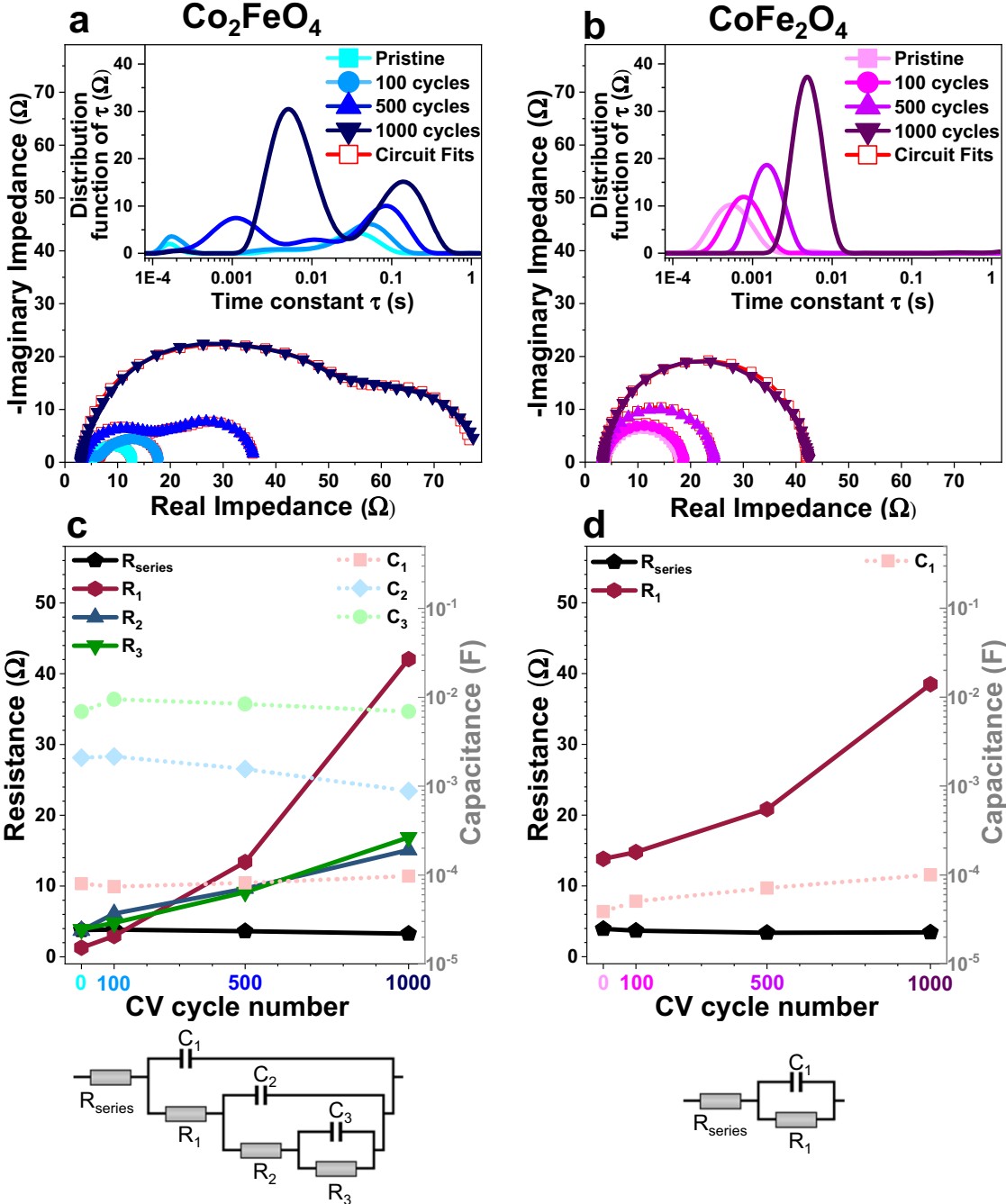

**Fig. 7 Electrochemical sub-processes of Co₂FeO₄ and CoFe₂O₄ during OER.** Electrochemical impedance spectroscopy data in complex plane representation (Nyquist plot) and determined distribution of relaxation times (insets) of electrodes covered by **a** Co₂FeO₄ nanoparticles at 1.63 V vs. RHE ($\approx E_{LSV,initial}$ (6 mA/cm²$_{geom}$)) and **b** CoFe₂O₄ nanoparticles at 1.73 V vs. RHE ($\approx E_{LSV,initial}$ (6 mA/cm²$_{geom}$)) in the pristine state and after 100, 500 and 1000 cycles. **c, d** Corresponding changes of the resistances (solid lines) and capacitances (dashed lines) as the number of CV cycles increases (obtained by equivalent circuit fitting to the displayed model circuits). Source data are provided as a Source Data file.

R₁, accompanied by a slightly decreasing C₂. This result suggests inhibited kinetics of adsorption of OH⁻ and an occurrence of irreversible oxidation of Co, which is consistent with our TEM and APT observation of structural transformation towards (Co, Fe)OOH and (Co, Fe)O₂ as the number of CV cycles increases. In contrast, no pseudocapacitive behaviour was observed for CoFe₂O₄ (Fig. 7b), which may relate to its low OER activity compared to Co₂FeO₄, as indicated by Tafel plots (Fig. 1e, f). For both Co₂FeO₄ and CoFe₂O₄, the increasing faradaic resistances (Fig. 7c, d) explain that the overpotential increases with the number of CV cycles (Fig. 1a, b), which is most likely arisen from

irreversible structural transformation towards inactive phases, as revealed by our TEM (Fig. 3c, f and Supplementary Figs. 2f and 3f) and APT data (Tables 1 and 2).

**Discussion**
Our study demonstrates that the deactivation process of Co₂FeO₄ and CoFe₂O₄ is closely associated with their structural and compositional evolution during OER, as schematically summarised in Fig. 8. Our APT data (Fig. 4i, m) unprecedentedly reveals 'segregated' Co₂FeO₄ whose compositional modulation is

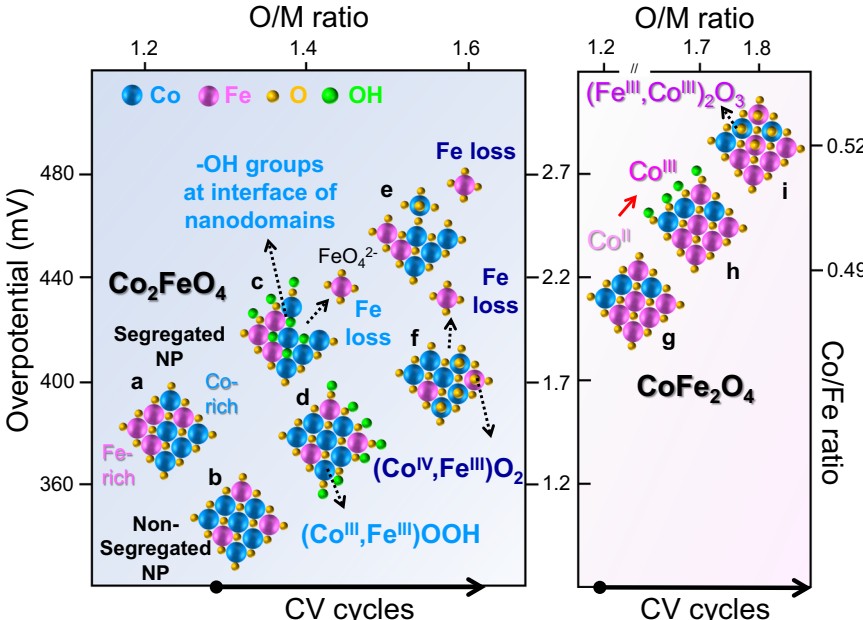

**Fig. 8 Schematic diagram of the gradual changes of Co₂FeO₄ and CoFe₂O₄ with increasing CV cycles under the OER conditions.** The pristine Co₂FeO₄ is comprised of **a** segregated Co₂FeO₄ with Fe-rich and Co-rich nanodomains and **b** non-segregated Co₂FeO₄ nanoparticles. The interface between Co-rich and Fe-rich nanodomains of segregated Co₂FeO₄ is thought to provide trapping sites for **c** hydroxyl groups, thereby contributing to a lower overpotential of pristine Co₂FeO₄ than that of **g** pristine CoFe₂O₄. At the beginning of electrolysis, concurrent structural transformation to **c, d** Co$^{III}$OOH and **c, e, f** Fe dissolution occurred for both segregated and non-segregated Co₂FeO₄. In addition, **f** stable (Co$^{IV}$, Fe$^{III}$)O₂ is formed on the surface of Co₂FeO₄, further degrading its activity. The OER activity of CoFe₂O₄ also decreases as the number of CV cycles increases, which is due to **h** oxidation of Co(II) to Co(III) and **i** the formation of the stable (Fe$^{III}$, Co$^{III}$)₂O₃ phase on the surface. (The left axis is overpotential at 10 μA/cm², the top axis is the average O/M ratio, the right axis is the average Co/Fe ratio, and the bottom arrows point to the trend of increasing CV cycles).

driven by spinodal decomposition[21,22], and 'non-segregated' Co₂FeO₄ in pristine Co₂FeO₄ (Fig. 8a, b). We speculate that such composition modulation is present in mixed Co$_x$Fe$_{(3-x)}$O₄ spinel oxides when x is in the range of 1.1–2.7 due to the miscibility gap. The interface between the Co-rich and Fe-rich nanodomains of 'segregated' Co₂FeO₄ (Fig. 4j, k) traps the hydroxyl groups, possibly due to the elastic strain induced by the difference in lattice constants (Fig. 8c). At the onset of OER, (Co$^{III}$, Fe$^{III}$)OOH is formed epitaxially on the surface of non-segregated Co₂FeO₄ (Fig. 3b and Supplementary Fig. 9a, b), and possibly in the nanodomains of segregated Co₂FeO₄ nanoparticles (Fig. 8d). Despite the formation of active (Co$^{III}$, Fe$^{III}$)OOH[13], the activity of Co₂FeO₄ decreases as a consequence of the irreversible structural transform towards (Co$^{IV}$, Fe$^{III}$)O₂ (Fig. 3c and Supplementary Fig. 2f) along with gradual loss of Fe (Tables 1 and 2, Figs. 4m, n and 5m, n) via the formation of soluble ferrate ions FeO₄²⁻ in alkaline electrolytes[13,43,56] (Fig. 8c, e–f). Therefore, we conclude that the concurrent structural transformation towards a stable (Co$^{IV}$, Fe$^{III}$)O₂ phase and Fe dissolution lead to a decrease in OER activity of Co₂FeO₄ as the number of CV cycles increases. For CoFe₂O₄, Co$^{II}$ present on the surface of pristine sample (Fig. 8g) is oxidised to Co$^{III}$ at the onset of OER (Fig. 8h), as shown by XPS data (Fig. 2b). After 1000 cycles, the (Fe$^{III}$, Co$^{III}$)₂O₃ phase, whose dimension is 4–5 nm, is likely formed on the surface of CoFe₂O₄ (Fig. 8i), as evidenced by both HRTEM and electron diffraction pattern (Fig. 3f and Supplementary Fig. 3f) and oxygen-rich regions revealed by APT (Fig. 6k, l, n), which slightly decreased the OER activity.

Importantly, our study demonstrates that OER is catalysed concurrently by multiple active regions in a single nanoparticle. Firstly, we observe that Co$^{II}$ catalyses the OER of both Co₂FeO₄ and CoFe₂O₄, since the Co oxidation state increases (indicated by XPS in Fig. 2 and XANES in Supplementary Fig. 7). The Co$^{II}$ in the

tetragonal sites (termed $Co_{Td}^{II}$) are thought to be active sites as they are responsible for the formation of active Co$^{III}$OOH species[17]; this is confirmed by our TEM data (Fig. 3b and Supplementary Fig. 9a, b) and the decreased number of tetrahedral sites by XANES data (inset, Supplementary Fig. 7a) as well as EIS data (Fig. 7a) after OER. Secondly, and most importantly, our results demonstrate that, Fe, when in the presence of $Co_{Td}^{II}$, plays a key role in the OER activity. Specifically, the amount of Fe loss is thought to indicate how rigorously OER occurs. Our APT data (Table 1) reveals that different regions have various levels of Fe dissolution, i.e., oxygen-rich regions of CoFe₂O₄ < oxygen-rich regions of non-segregated Co₂FeO₄ ≤ Fe-rich nanodomains in segregated Co₂FeO₄ < Co-rich nanodomains in segregated Co₂FeO₄. The number of $Co_{Td}^{II}$ sites in these regions also increases in a similar manner (as indicated by Supplementary Table 3, estimated according to refs. [57,58]). These observations provide a strong indication that Fe, which can actively catalyse OER, seems to be associated with the number of $Co_{Td}^{II}$ sites. Therefore, we hypothesis that the presence of Co oxyhydroxide is imperative for the activation of Fe, since FeOOH has low electrical conductivity and stability at lower potentials[56], and Co$^{III}$OOH provides an electrically conductive support[9]. A previous study reported that Fe can serve as an indirect active site, for example by changing the spin and charge state of Co[11], or by assisting[5] or stabilising the active Co$^{III}$OOH intermediate[59], thereby enhancing OER activity. Another study proposed that di-μ-oxo bridged Co-Fe sites act as active sites above a transition voltage, below which di-μ-oxo bridged Co-Co sites catalyse OER[4]. Our study indicates that Fe promotes the formation of active species for OER, possibly Co$^{III}$Fe$^{III}$OOH, whose OER activity is significantly better than that of Co$^{III}$OOH[13] (although its activity drops rapidly due to Fe dissolution). Therefore, in comparison to CoFe₂O₄, Co₂FeO₄ has a higher OER activity most likely due to the formation of more Co$^{III}$Fe$^{III}$OOH yielded by its optimal ratio of Fe content to $Co_{Td}^{II}$ sites. Also, the interface between

the nanodomains of segregated $Co_2FeO_4$ traps hydroxyl groups, providing additional active regions for OER, thereby further enhancing the OER activity of $Co_2FeO_4$. Therefore, we conclude that the presence of $Co^{III}Fe^{III}OOH$, promoted by Fe and $Co_{Td}^{II}$, coupled with the nanosized defect features, leads to $Co_2FeO_4$ having an increased OER activity. This potentially explains how the addition of a small amount of Fe improves the OER activity of mixed Co-Fe spinel oxides[7,8].

In summary, our study provides atomic-scale insights into the evolving surface structure of $Co_2FeO_4$ and $CoFe_2O_4$ nanoparticles during OER and reveals how those structural and compositional changes alter the activity and stability. We demonstrate the importance of 3D atomic-scale imaging and quantitative compositional analysis of nanoparticles in both their pristine state and at various stages of electrochemical reaction when seeking to understand their activity and stability. We believe that APT, when combined with X-ray- and electron-based characterisation techniques, has enormous potential to better understand the reaction and degradation mechanisms of oxide or metallic catalyst nanoparticles during important reactions, such as OER and $CO_2$ reduction.

## Methods

**Synthesis of nanoparticles**. Iron (III) nitrate (Fe $(NO_3)_3 \cdot 9H_2O$), cobalt (II) nitrate ($Co(NO_3)_2 \cdot 6H_2O$), ammonia ($NH_3 \cdot H_2O$) and polyethene glycol (PEG, $Mn = 400$) were purchased from Sinopharm Chemical Reagent (Shanghai, China). The $CoFe_2O_4$ nanoparticles were prepared by dissolving 3.2 g of $Fe(NO_3)_3 \cdot 9H_2O$, 1.2 g of $Co(NO_3)_2 \cdot 6H_2O$ and 0.2 g of polyethene glycol (PEG, Mn = 400) in 40 mL of ultrapure water (0.055 $\mu S\ cm^{-1}$) under vigorous stirring for 30 min[60]. For $Co_2FeO_4$ nanoparticles, 1.6 g of $Fe(NO_3)_3 \cdot 9H_2O$, 2.3 g of $Co(NO_3)_2 \cdot 6H_2O$ and 0.2 g of PEG were mixed in 40 mL of ultrapure water. 5 mL of ammonia diluted with 5 mL of ultrapure water was slowly added dropwise into the solution mixture. The obtained suspension was subsequently transferred into a 100 mL Teflon-lined stainless autoclave, maintained at 180 °C for 3 h. The product was washed several times with ultrapure water via centrifugation, dried in an oven at 80 °C for 12 h.

**XRD measurements**. The XRD analysis of the pristine $Co_2FeO_4$ and $CoFe_2O_4$ nanoparticles was performed on an Rigaku Ultima IV diffractometer with Cu Kα radiation ($\lambda = 0.15418$ nm) at a scanning speed of 4°/min, scanning step of 0.02° and operating voltage of 40 kV. The XRD data was given in Supplementary Fig. 1.

**TEM measurements**. TEM and HRTEM experiments were carried out in an aberration-corrected JEOL JEM-2200FS operating at 200 kV, and TEM/EDX data was acquired with an Oxford X-max detector. The TEM samples were prepared by dispersing a small amount of nanoparticles into anhydrous ethanol via ultrasonication, followed by dropping nanoparticle solutions on Cu TEM grids and dried at room temperature. The HRTEM images were processed by using Gatan Digital Micrograph. Additional TEM images were shown in Supplementary Figs. 2, 3, and 8–10.

**Electrochemical measurements**. Electrochemical measurements were performed in a three-electrode system at an electrochemical workstation (PalmSens3), where a Pt wire and Ag/AgCl (3 M KCl) served as counter and reference electrodes. The OER performance was studied by using a rotating disk electrode (10 mm diameter, 0.785 $cm^2$). The glassy carbon electrode was first cleaned and polished to a mirror finish with 50 nm $Al_2O_3$. In all, 32 μL of dispersion was transferred onto the glassy carbon disk and then dried at room temperature. The dispersion was prepared by dispersing 5 mg of nanoparticle powder in 1 mL ultrapure water, followed by ultrasonication for 30 min. The LSV curves were recorded with a scan rate of 10 mV/s in 1.0 M KOH solution from 0 V to 0.8 V (vs Ag/AgCl) at a rotating speed of 1600 rpm. The CV measurements were performed at a scan rate of 50 mV/s from 0 V to 0.65 V (vs Ag/AgCl). Electrochemical data normalised to electrode geometric surface area was provided in Supplementary Fig. 5. Another set of electrochemical measurements was carried out at the same conditions except for the KOD solution in $D_2O$ for APT specimen preparation and measurements. The deuterium oxide ($D_2O$ with 99.9 at.% D) and potassium deuteroxide solution (40 wt.% KOD in $D_2O$ with 98 at.% D) were purchased from Sigma-Aldrich. EIS was performed under OER conditions by applying a sine wave signal with a 10 mV amplitude in the frequency range from 6 kHz to 0.2 Hz after equilibrating 5 s at 0.6 V vs. Ag/AgCl for $Co_2FeO_4$ and 0.7 V vs. Ag/AgCl for $CoFe_2O_4$. The software RelaxIS 3 (rhd instruments) was used for data treatment and analysis (details were given in Supplementary Note 4, Supplementary Figs. 24 and 25).

**$N_2$ Physisorption measurements**. The $N_2$ sorption experiments were carried out in a physical adsorption analyser (TriStar II 3020, Micromeritics). In all, 87 mg $CoFe_2O_4$ and 72 mg $Co_2FeO_4$ nanoparticles were degassed at 150 °C for 8 h under

vacuum before measurement. The BET surface area was calculated within the relative pressure range of 0.05–0.3 ($p/p°$). Data was shown in Supplementary Fig. 4.

**XPS measurements**. XPS measurements were performed on a VersaProbe II (Ulvac-Phi) using a monochromatic Al X-ray source (1486.6 eV) operating at 15 kV and 13.2 W. The emission angle between the analyser and the substrate surface is 45°. The binding energy scale was referenced to the main C 1 s signal at 284.8 eV. Detailed Analysis of the spectra was carried out with the software CasaXPS. Peak fitting was revealed in Supplementary Fig. 6.

**XAS measurements**. Co Kβ High Energy Resolution Fluorescence Detected (HERFD) XAS and Co Kβ XES were collected at beamline I20 at the Diamond Light Source (3 GeV, 300 mA). A Si (111) double crystal monochromator was used for energy selection of the incident beam, and a rhodium-coated mirror was used for harmonic rejection, delivering a flux of $\sim 4 \times 10^{12}$ photons/s at the sample position. X-rays were focused to achieve an approximate beam size of $100 \times 300\ \mu m^2$ (VxH). A Johann-type XES spectrometer was used with two Ge (444) crystals aligned by setting the maximum of the Kβ emission line of a Co foil to 7059.1 eV. The incident energy was calibrated by setting the first inflection point of the Co XAS spectra to 7709.0 eV for a Co foil. Co Kβ XES spectra were collected from 7620 to 7670 eV, with a step size of 0.2 eV. The HERFD XAS edge spectra were collected with the spectrometer fixed at the maximum of the Kβ emission energy while scanning the energy of the incident monochromator. Co Kβ-detected HERFD XAS spectra were collected from 7690 to 7745 eV, with a step size of 0.25 eV over the edge region (7690−7725 eV) and steps of 0.5 eV over the EXAFS region (7725−8500 eV). Co and Fe K-edge XAS spectra were collected in fluorescence mode using a 4-element Vortex Si-drift detector for all samples but the Co Kb detected HERFD for $CoFe_2O_4$. Pre-edge baseline corrections were done using Larch XAS Viewer[61]. XAS data are discussed in Supplementary Note 1 and shown in Supplementary Fig. 7.

**APT measurements**. Before preparing the needle-shaped APT specimen, a bulk material containing nanoparticles was fabricated by the following procedure. A nanoparticle suspension was prepared by mixing 5.0 mg of nanoparticles in 1.0 ml of $D_2O$ water, followed by ultrasonication for 30 min. 20 μL of nanoparticle solutions were dropped on a clean glassy carbon and dried at room temperature overnight. The CV measurements were performed on the glassy carbon electrodes in 1 M KOD (in $D_2O$) solution at a scan rate of 50 mV/s from 0 V to 0.65 V (vs Ag/AgCl) for 100, 500 and 1000 cycles, respectively. Afterwards, the glassy carbon electrode was covered by Ni electrodeposition at a constant voltage of −1.5 V for 300 s in an electrolyte mixed with 3.0 g nickel sulphate, 0.5 g nickel chloride, and 0.5 g boric acid in 10 ml DI water[62]. This bulk material was subsequently used to prepare needle-shaped APT specimens by a lift-out procedure by a focus ion beam/scanning electron microscope (FEI Helios G4 CX) (details are shown in Supplementary Note 2 and Supplementary Fig. 11). The APT experiments were conducted in CAMECA LEAP 5000XR instrument in laser pulsing mode at a specimen temperature of 57 K, laser energy of 30 pJ, pulse frequency of 125 kHz, and detection rates of 0.5. The APT data are reconstructed and analysed using the commercial IVAS 3.8.2 software. Additional APT data and analysis were listed in Supplementary Figs. 12–22 and Supplementary Note 3.

**$H_2$ TPR measurements**. The $H_2$ TPR measurements were conducted in a flow set-up consisting of a gas supply, a stainless-steel U-tube reactor heated in a ceramic tube furnace, and a thermal conductivity detector (TCD, Hydros 100). For the measurement, 116.6 mg catalyst nanoparticles were pre-treated in 50 Nml $min^{-1}$ He (99.9999%) at 400 °C for 1 h. After cooling to 60 °C, the set-up was flushed with 50 Nml $min^{-1}$ 4.58% $H_2$ (99.9999%)/Ar (99.9995%) for 1 h. The furnace was heated to 800 °C with a heating rate of 10 K $min^{-1}$. The maximum temperature was kept constant for 1 h. The temperature of the sample was measured every two seconds by a thermocouple placed inside the reactor. The arising water was condensed in a cold trap. The measured consumption of $H_2$ and temperature were plotted against the measurement time. Experimental data was shown in Supplementary Fig. 23 and Supplementary Note 3.

## Data availability

The raw datasets generated and/or analysed during the current study are available in Figshare[63]. Source data are provided with this paper.

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

## Acknowledgements

W.X. and T.L. thank Deutsche Forschungsgemeinsschaft (DFG) for financial support (project number 407513992). T.F., M.M., J.L., K.T., O.R., S.D., T.L., U.H. and M.H. thank DFG (Projektnummer 388390466 – TRR 247/ A1, A2, B6, C4 and S projects) for financial support. W.X. and T.L. would like to thank Zentrum für grenzflächendominierte Höchstleistungswerkstoffe (ZGH) at Ruhr University Bochum for the access to infrastructure (FEI Helios G4 CX SEM/FIB and Cameca LEAP 5000 XR). O.R. and S.D. thank the Max Planck Society for support. J.L. and K.T. acknowledge the Center for Solvation Science (ZEMOS). T.L. thanks Mr. Arjun BalaKrishnan and Mr. Naiyu Qi for assistance with processing figures.

## Author contributions

T.L. designed, supervised and coordinated the project. N.Y. and X.L. synthesised the nanoparticles and conducted the XRD and BET measurements. W.X. performed the electrochemical measurements, J.L. and K.T. contributed to data analysis and interpretation. U.H. performed the XPS measurements and analysed the XPS data. O.R., M.A. and S.D. performed XAS measurements and data analysis. M.H. performed the TEM measurements and T.L. analysed the TEM data. T.F. and M.M. performed the $H_2$/TPR experiment. W.X. performed the APT experiments, W.X. and T.L. analysed the APT data. All authors agreed on the contents and conclusion of the paper.

## Funding

## Competing interests

The authors declare no competing interests.
