## [Peer Review File · Nature Communications]

3D atomic-scale imaging of mixed Co-Fe spinel oxide nanoparticles during oxygen evolution reactionREVIEWER COMMENTS

Reviewer #1 (Remarks to the Author):

Dear Authors,

The manuscript titled “3D atomic-scale imaging of mixed Co-Fe spinel oxide nanoparticles during oxygen evolution reaction” has been objectively reviewed. Overall, the manuscript is very well written with no major issues about the technical aspects. The paper presents an impressive amount of well analyzed data from a variety of complementary analytical techniques, from which the authors draw links between the evolution of catalyst electrochemical efficiency to observed changes with the atomic to nanoscale structural and compositional heterogeneity of individual nanoparticles. The authors impressively overcame many technical challenges, including the APT analysis of many individual real catalyst nanoparticles, to convincingly make logical conclusions based on clear results and show how the application of APT is (and will continue to be) an essential and unique tool necessary to uncovering the atomic-scale dynamics of the OER. Additionally, the results address some controversy in the field regarding the role of Fe in mixed Co-Fe oxides for catalyzing oxygen evolution reactions, as well as bring to light other aspects that significantly advance the field such as demonstrating that OER is catalyzed concurrently by multiple active regions in a single nanoparticle.

It is thus my opinion that this manuscript overall is suitable for publication in Nat. Comm. as it makes it significant contribution to the field. However, before it should be considered for acceptance, I ask that the following be addressed:

1. Please change the “green” color of the O (OD) spheres of the APT data. The color contrast compared to the blue and magenta CoO_x and FeO_y species makes it hard to distinguish OD. (e.g. Fig. 4e-h).
2. Based on the shape of distributed species, the selected zoomed in APT maps in Fig. 4e-h do not appear to be from their respective highlighted volumes shown by the dashed boxes in Fig. 4a-d; see 4b versus what is shown in 4f as a clear example. Please address
3. How representative, are the HRTEM measurements with regard to what is presented? From lines 249: “Our detailed APT analysis of 48 pristine Co₂FeO₄ nanoparticles (Fig. 4a and Supplementary Fig. 10a) reveals that 26 of them have nanoscale compositional modulation, while the remaining 22 exhibit a relatively uniform elemental distribution”. This shows good representation (statistics) from APT. Although TEM-based size distribution is presented (Supplementary Fig 2 and 3), there is no mention of the statistics for structure determination. Please address.

4. Clearly the corresponding author is an expert with APT. Please provide some comments/statements regarding the measurement of stoichiometry especially around the O considering issues that can arise (e.g. Phys Chem Lett, 2013, 4, 993-998).

5. Regarding Supplementary Note 3 – APT specimen preparation: it is mentioned that the “electrodeposited Ni layer consists of a good distribution of nanoparticles. Can you provide details as to how this could be considering the description of the NP deposition onto the glassy carbon electrode? It is not clear how Ni would envelope NPs adsorbed on the electrode surface unless there is free floating in the Ni plating solution. Please provide details. Additionally, is the Ni plating procedure following (or at least inspired by) work by Kim et al. “A New Method for Mapping the Three-Dimensional Atomic Distribution within Nanoparticles by Atom Probe Tomography (APT).” Ultramicroscopy, 2018, 190, 30–38. If so, please reference this work or previous published work by others.

Reviewer #2 (Remarks to the Author):

The authors present a study that combines X-ray characterization, high-resolution electron microscopy and atom probe tomography (APT) to study structural and chemical changes that occur to CoFe₂O₄ and Co₂FeO₄ electrocatalysts during the anodic oxygen evolution reaction. Similar surface changes have been reported for CoFe₂O₄ and Co₂FeO₄ electrocatalysts during OER using in situ X-ray spectroscopy, and the current XPS and XANES data is consistent with data in J. Mater. Chem. A, 2018, 6, 7034 (DOI: 10.1039/c7ta10892c). The novel aspect of this manuscript is the atomic-level resolution the authors obtained using APT to identify specific locations of adsorbed hydroxide species and nanometer-scale heterogeneity within individual catalyst particles. In particular, the authors used this information to correlate composition-dependent deactivation pathways with the resulting end-of-run surface structures for both CoFe₂O₄ and Co₂FeO₄.

The results are well presented, but the findings that localized heterogeneity and phase separation occur during OER is not unexpected. For example, it is well known that CoFeOx and NiFeOx catalysts will experience phase changes, (oxy)hydroxide formation, and potential Fe dissolution (or deposition from solution) during OER. However, the nm-scale spatial resolution of their analytical technique does provide new insight into the deactivation processes of CoFeOx-based OER catalysts and should interest the broader electrocatalyst community.

The authors should address the following points before this manuscript is further considered for publication.

(1) The XANES data is largely consistent with previously published results for Co₂FeO₄ and CoFe₂O₄ (J. Mater. Chem. A, 2018, 6, 7034; DOI: 10.1039/c7ta10892c). The authors should cite this paper when explaining the spectral differences.

(2) The above noted paper CoFe₂O₄ to be much more stable than Co₂FeO₄ after 1000 CV cycles to +1.8V vs. RHE in 0.1M KOH. In the present manuscript both catalysts showed severe deactivation. The authors should comment on this apparent discrepancy with previous literature.

(3) The main text and experimental section state that 1.0M KOH was used as an electrolyte, but the Figure 1 legend states 0.1M KOH. The authors should confirm the electrolyte concentration.

(4) Supp. Figure 4 lists the BET-derived specific surface of the catalysts with a unit of m²/g². Should this be m²/g?

(5) The catalyst current density is normalized to BET-derived surface area. The authors should also report the geometric current density normalized to the electrodes geometric area.

(6) Page 4, line 120: authors report a Tafel slope of “ $\sim 83 \pm 1.7$ mV/dec” for Co₂FeO₄ after 1000 OER cycles. Why does this value have an approximate (\sim) designation since it has a reported standard deviation?

Reviewer #3 (Remarks to the Author):

This paper reports microstructure investigation of Co-Fe spinel oxides using APT method: it is possibly suitable for publication in this journal, but the authors should correct their lack of clarity.

The manuscript shows abundant observations including X-ray spectra and TEM images.

It also demonstrates multiscale investigations, combining chemical states at surface region by XPS, local co-ordinations by XAFS, phase separations by APT and etc.

The authors should mention the observing scales and regions of each technique briefly in the introduction or method sections.

Such informations might help readers to understand the roles of techniques and the relations of their results.

Response to Reviewers' comments on Nature Communications Manuscript (NCOMMS-21-33583)

We thank the Reviewers for the constructive input. Below are point-by-point responses to the issues raised by the Reviewers, all of which are incorporated in the attached manuscript. The Reviewers' comments are in dark blue fonts and our responses in black fonts. We have also highlighted the revised text in red in the main text.

Reviewer #1 (Remarks to the Author):

Dear Authors,

The manuscript titled "3D atomic-scale imaging of mixed Co-Fe spinel oxide nanoparticles during oxygen evolution reaction" has been objectively reviewed. Overall, the manuscript is very well written with no major issues about the technical aspects. The paper presents an impressive amount of well analysed data from a variety of complementary analytical techniques, from which the authors draw links between the evolution of catalyst electrochemical efficiency to observed changes with the atomic to nanoscale structural and compositional heterogeneity of individual nanoparticles. The authors impressively overcame many technical challenges, including the APT analysis of many individual real catalyst nanoparticles, to convincingly make logical conclusions based on clear results and show how the application of APT is (and will continue to be) an essential and unique tool necessary to uncovering the atomic-scale dynamics of the OER. Additionally, the results address some controversy in the field regarding the role of Fe in mixed Co-Fe oxides for catalysing oxygen evolution reactions, as well as bring to light other aspects that significantly advance the field such as demonstrating that OER is catalysed concurrently by multiple active regions in a single nanoparticle.

We appreciate Reviewer's compliment.

It is thus my opinion that this manuscript overall is suitable for publication in Nat. Comm. as it makes it significant contribution to the field. However, before it should be considered for acceptance, I ask that the following be addressed:

1. Please change the "green" color of the O (OD) spheres of the APT data. The color contrast compared to the blue and magenta CoOx and FeOy \rightarrow species makes it hard to distinguish OD. (e.g. Fig. 4e-h).

We thank Reviewer's suggestion. We have changed the color of OH(D) spheres to dark blue. We have updated the atom maps in Figs. 4-6 and Supplementary Figs. 15, 17 and 20, and modified the text in page 13:

‘Importantly, we observed that the hydroxyl groups, shown as dark blue spheres in 2D Fe compositional profiles of Figs. 4j-l’.

- Based on the shape of distributed species, the selected zoomed in APT maps in Fig. 4e-h do not appear to be from their respective highlighted volumes shown by the dashed boxes in Fig. 4a-d; see 4b versus what is shown in 4f as a clear example. Please address

We have changed the atom maps in Figs. 4a, c-d, so that the viewing angles of nanoparticles in the dashed boxes are consistent with those nanoparticles shown in Figs. 4e, g-h (see updated Fig. 4 below). The atom map of Fig. 4f is obtained from the X-Y view of Fig. 4b, while Fig. 4b is shown in the X-Z view. This is clearly illustrated in Figure R1. We displayed the nanoparticle in Fig. 4f in the X-Y view because it gives us the best ‘edge-on’ compositional contrast of the Co-rich and Fe-rich regions in the 2D Fe composition map (Fig. 4j), since the nanoparticle is ~45 degrees tilted to the X-Z or Y-Z plane. We could replace Fig. 4b with Figure R1b. But this is the top-down view atom map, which would seem inconsistent with other side-view atom maps (X-Z or Y-Z) shown in Figs. 4a, c-d. Therefore, we keep the atom map in Fig. 4b but clarify this point in the caption (see the red-color text).

Fig. 4 Compositional evolution and elemental distribution of segregated Co_2FeO_4 during OER. (a-d) 3D-APT reconstruction of Co_2FeO_4 in the pristine state and after 100, 500 and 1000 cycles of CV showing the nanoparticles embedded in Ni matrix, (e-h) 3D atom maps of segregated Co_2FeO_4 nanoparticles in the pristine state and after 100, 500 and 1000 cycles, selected from dashed line region in Fig. 4a-d (f displayed in the top-down view of b), and (i-l) 2D Fe compositional maps of the same nanoparticles in (e-h) showing the Co-rich and Fe-rich nanodomains.

Figure R1. (a) 3D-APT reconstruction (X-Z, side view) shown in Fig. 4b, and (b) X-Y (top-down) view of the same reconstruction.

3. How representative, are the HRTEM measurements with regard to what is presented? From lines 249: “Our detailed APT analysis of 48 pristine Co_2FeO_4 nanoparticles (Fig. 4a and Supplementary Fig. 10a) reveals that 26 of them have nanoscale compositional modulation, while the remaining 22 exhibit a relatively uniform elemental distribution”. This shows good representation (statistics) from APT. Although TEM-based size distribution is presented (Supplementary Fig 2 and 3), there is no mention of the statistics for structure determination. Please address.

We have added two more HRTEM images of Co_2FeO_4 and CoFe_2O_4 nanoparticles for each condition to show similar structural changes observed in Fig. 3 (see new Supplementary Fig. 9 below). Additionally, we confirmed the structural changes of Co_2FeO_4 and CoFe_2O_4 after 1000 cycles by their corresponding selected area electron diffraction (SAED) patterns, Figures R2b, 2d. The SAED pattern was recorded in a 500 nm X 500 nm region containing more than 100 nanoparticles. We observed, in Figure R2b, additional reflection spots by comparing the SAED pattern of the pristine Co_2FeO_4 (Figure R2a); the extra reflection spots correspond well to $(11\bar{2}0)$ CoO_2 . Also, extra reflection spots were observed for the CoFe_2O_4 nanoparticles after 1000 cycles, suggesting the presence of $(30\bar{3}0)\text{Fe}_2\text{O}_3$ or $(202)\text{FeOOH}$. As mentioned in page 8, ‘According to the Fe Pourbaix diagram [43], Fe_2O_3 forms as the potential increases. Thus, the surface of inverse spinel CoFe_2O_4 is thought to undergo a phase transformation to the $(\text{Fe, Co})_2\text{O}_3$ phase after 1000 cycles.’, we thus ascribed the additional reflection spots to $(30\bar{3}0)\text{Fe}_2\text{O}_3$. Hence, it is statistically enough to confirm the structural transformation of Co_2FeO_4 and CoFe_2O_4 nanoparticles after OER based on the newly added HRTEM data (Supplementary Fig. 9) and SAED patterns (incorporated in Supplementary Figs. 2f and 3f).

We have also modified the text in pages 7-8:

‘After 100 cycles, a distinct structural change occurs on the surface of Co_2FeO_4 , as highlighted by the blue-dotted areas in Fig. 3b (more HRTEM images shown in Supplementary Fig. 9).

‘The formation of CoO_2 is also confirmed by the observation of additional reflection spots that correspond well to (11-20) CoO_2 in the selected area electron diffraction (SAED) pattern after 1000 cycles (Supplementary Fig. 2f) compared to that of the pristine Co_2FeO_4 (Supplementary Fig. 2e); the SAED pattern was recorded from a 500 nm x 500 nm area containing more than 100 nanoparticles.’

‘Also, the SAED pattern of CoFe_2O_4 , in Supplementary Fig. 3f, reveals additional reflection spots after 1000 cycles, indicating the presence of (30-30) Fe_2O_3 or (202) FeOOH . According to the Fe Pourbaix diagram [43], Fe_2O_3 forms as the potential increases. Thus, the surface of inverse spinel CoFe_2O_4 is thought to undergo a phase transformation to the $(\text{Fe, Co})_2\text{O}_3$ phase after 1000 cycles.’

Figure R2. Selected area electron diffraction patterns of Co₂FeO₄ and CoFe₂O₄ in the (a, c) pristine state and (b,d) after 1000 cycles of cyclic voltammetry measurements.

4. Clearly the corresponding author is an expert with APT. Please provide some comments/statements regarding the measurement of stoichiometry especially around the O considering issues that can arise (e.g. Phys Chem Lett, 2013, 4, 993-998).

Inspired by the previous study (Phys Chem Lett, 2013, 4, 993-998), we have investigated the effect of laser pulse energy on the measurement of oxide stoichiometry of Co₂FeO₄ nanoparticles by APT (see Supplementary Note 4). The APT specimens were analysed at voltage pulsing mode with a pulse fraction of 20%, and at laser pulsing mode with laser energy of 10, 30 and 60 pJ, respectively (all at a specimen temperature of 57 K, pulse frequency of 125 kHz and detection rate of 0.5). We observed that the oxygen concentration measured at voltage pulsing mode is 40.1 ± 0.4 at.% (Supplementary Fig. 22c). The value increases with laser energy until it reaches a plateau of ~ 46.0 at.% at 30 pJ (Supplementary Fig. 22c). This

result indicates an oxygen deficiency regardless of voltage or laser pulsing mode. To investigate this, we have further analysed the charge-to-state ratios. The ratios of $\text{Co}^{1+}/\text{Co}^{2+}$ and $\text{Fe}^{1+}/\text{Fe}^{2+}$ increase as the laser energy increases (Supplementary Fig. 22a), suggesting that higher laser energy yields less ionisation of secondary ions as the electric field decreases. The $\text{O}_2^{1+}/\text{O}^{1+}$ ratio increases with the laser energy, indicating an enhanced complex ion generation at higher laser energy (Supplementary Fig. 22b). Additionally, the ratio of multiple hits to single hits at 16 Da was plotted in Supplementary Fig. 22d. The multiple hits/single hits ratio is 13.2:1 at voltage pulsing mode, and it drops significantly and reaches a plateau of 1.9:1 after 10 pJ. We speculate that the multiple hits, i.e., $^{16}\text{O}_2^{2+}$ at 16 Da, lead to the oxygen deficiency since the $^{16}\text{O}_2^{2+}$ complex ions most likely evaporate as multiple hits, similar to $^{12}\text{C}_2^{2+}$ observed for carbides (Ultramicroscopy, 2013, 132, 239-247). Additionally, the proportion of multiple hits at 16 Da decreases as the electric field is lowered, possibly explaining why the oxygen concentration increases with laser energy.

Also, we want to point out that our result shows an opposite trend to the previous study (Phys Chem Lett, 2013, 4, 993-998). One key difference is that bulk MgO was analysed in the previous study, while in our study the oxide nanoparticles (8-10 nm in dia.) embedded in Ni were analysed. Essentially, the bulk part of our APT specimens is metal. We speculate that the difference in oxides' volume fraction (or location) might induce different field evaporation mechanisms during laser pulses. The previous study (Phys Chem Lett, 2013, 4, 993-998) provides convincing evidence that the oxygen deficiency is induced by formation of oxygen neutrals triggered by an electron excitation (photoexcitation) of bulk oxides under laser pulses. In comparison to our study, such electronic excitation might not be the dominant field evaporation mechanism since the bulk part of our APT specimens is metallic Ni that should evaporate as ions or complex molecular ions induced by thermal-assisted field evaporation. Therefore, we speculate that the multiple hits at 16 Da is the dominant reason for the oxygen deficiency in our work.

Furthermore, to solve this issue, we verified the oxygen content measured by APT with the actual oxygen content measured by the H_2 -temperature programmed reduction (H_2 TPR) measurements (which is able to provide a nominal oxygen content). A reference measurement was first performed on CuO. Specifically, 0.1069 g of Cu was heated to 450 °C with a heating rate of 5 °C min^{-1} in 84.1 Nml min^{-1} 4.58 % H_2/Ar . We can calculate the ratio of oxygen to metal (O/Metal) by using the following equation:

$$\frac{O}{\text{Metal}} = \frac{n_O}{n_{\text{Metal}}} = \frac{n_O \times M_{\text{metal}}}{m - n_O \times M_O}$$

Where n_O and n_{Metal} are the molar quantity of oxygen and metal, M_O is the molar mass of oxygen, m is the weight of metal oxide sample. The consumption of H_2 (n_{H_2} , mol) is consistent with the O atom quantity ($n_O = n_{\text{H}_2}$) in the measured sample, calculated by the integral of the H_2 TPR curve (grey curve in Supplementary Fig. 23). The weight of the measured sample and total consumption of H_2 is 0.1069 g and 0.00135 mol, respectively. Thus, the O/Cu atomic ratio is 1.01, suggesting an oxide stoichiometry of $\text{CuO}_{1.01}$ for CuO. Afterwards, we performed H_2 TPR measurement on the pristine Co_2FeO_4 nanoparticles. The weight of Co_2FeO_4 and H_2

consumption is 0.1166 g and 0.00182 mol, respectively (blue curve in Supplementary Fig. 23). The molar mass of Co_2FeO_4 is estimated to be ~ 57.7 g/mol, which yields the $\text{O}/(\text{Co}+\text{Fe})$ ratio of 1.2 (the ratio of Co/Fe (~ 1.38) can be obtained from APT data). The $\text{O}/(\text{Co}+\text{Fe})$ ratio measured by APT is ~ 0.82 , 1.46 times lower than the value measured by H_2 TPR. In this study, we used this correction factor (1.46) to 'calibrate' the oxygen counts, as the proportion of multiple hits ($^{16}\text{O}_2^{2+}$) that causes the deficiency in oxygen content shall be similar when the same operating conditions were used for APT measurements.

To mention these details in Supplementary Note 4, we have added a sentence in page 10 and cite the previous work (Phys Chem Lett, 2013, 4, 993-998):

'Also, the effect of laser pulse energy on the measurement of oxide stoichiometry [48] was investigated and detailed in Supplementary Note 4.'

Supplementary Figure 22. The effect of laser energy on the (a) M^{1+}/M^{2+} ratio, (b) $\text{O}_2^{1+}/\text{O}^{1+}$ ratio, (c) O concentration, and (d) the ratio of multiple hits to single hits at 16 Da measured from APT data of pristine Co_2FeO_4 .

Supplementary Figure 23. H₂ TPR profiles of standard CuO and the pristine Co₂FeO₄ and nanoparticles.

- Regarding Supplementary Note 3 – APT specimen preparation: it is mentioned that the “electrodeposited Ni layer consists of a good distribution of nanoparticles. Can you provide details as to how this could be considering the description of the NP deposition onto the glassy carbon electrode? It is not clear how Ni would envelope NPs absorbed on the electrode surface unless there is free floating in the Ni plating solution. Please provide details. Additionally, is the Ni plating procedure following (or at least inspired by) work by Kim et al. “A New Method for Mapping the Three-Dimensional Atomic Distribution within Nanoparticles by Atom Probe Tomography (APT).” *Ultramicroscopy*, 2018, 190, 30–38. If so, please reference this work or previous published work by others.

We did not dissolve the post-OER nanoparticles in a suspension. Instead, we directly covered the glassy carbon electrodes, deposited with nanoparticles, by Ni capping via electrodeposition. Regarding the reason for a good distribution of nanoparticles embedded in the matrix (observed in Supplementary Fig. 11b), we speculate that the post-OER nanoparticles deposited on the glassy carbon electrode are possibly ‘re-arranged’ during electrodeposition. That is, the nanoparticles loosely bound on the Ni plate are detached and diffused into the electrolyte (near the cathode) at the onset of electrodeposition. Subsequently, these nanoparticles might be redeposited on glassy carbon concurrently with the electrodeposition of Ni. The diffusion coefficient of Ni ions in an aqueous solution is thought to be higher than that for nanoparticles. Thus, the deposition rate of nanoparticles might be slower than Ni. This possibly explains why the nanoparticles are well separated by Ni. We speculate that the distance between nanoparticles might depend on the local concentration of nanoparticles (near the cathode) and the difference in the diffusion rates between nanoparticles and Ni ions under the electric field.

We have cited the work (*Ultramicroscopy*, 2018, 190, 30–38) as Ref. [62] in the experimental section in page 23.

Reviewer #2 (Remarks to the Author):

The authors present a study that combines X-ray characterisation, high-resolution electron microscopy and atom probe tomography (APT) to study structural and chemical changes that occur to CoFe₂O₄ and Co₂FeO₄ electrocatalysts during the anodic oxygen evolution reaction. Similar surface changes have been reported for CoFe₂O₄ and Co₂FeO₄ electrocatalysts during OER using in situ X-ray spectroscopy, and the current XPS and XANES data is consistent with data in J. Mater. Chem. A, 2018, 6, 7034 (DOI: 10.1039/c7ta10892c). The novel aspect of this manuscript is the atomic-level resolution the authors obtained using APT to identify specific locations of adsorbed hydroxide species and nanometer-scale heterogeneity within individual catalyst particles. In particular, the authors used this information to correlate composition-dependent deactivation pathways with the resulting end-of-run surface structures for both CoFe₂O₄ and Co₂FeO₄.

The results are well presented, but the findings that localised heterogeneity and phase separation occur during OER is not unexpected. For example, it is well known that CoFeOx and NiFeOx catalysts will experience phase changes, (oxy)hydroxide formation, and potential Fe dissolution (or deposition from solution) during OER. However, the nm-scale spatial resolution of their analytical technique does provide new insight into the deactivation processes of CoFeOx-based OER catalysts and should interest the broader electrocatalyst community.

We thank Reviewer's comment on the novelty and significance of our work.

The authors should address the following points before this manuscript is further considered for publication.

- (1) The XANES data is largely consistent with previously published results for Co₂FeO₄ and CoFe₂O₄ (J. Mater. Chem. A, 2018, 6, 7034; DOI: 10.1039/c7ta10892c). The authors should cite this paper when explaining the spectral differences.

When we described the spectral differences, we have mentioned in page 7 that '*We observed a subtle shift of Co K-edge towards higher energy values (Supplementary Fig. 6a), possibly suggesting that only a small volume fraction, potentially on the surface regions, has an increase in oxidation state to Co(IV), which is in agreement with the XAS data of Ref. [20].*' Ref. 20 is the reference mentioned by the Reviewer.

- (2) The above noted paper CoFe₂O₄ to be much more stable than Co₂FeO₄ after 1000 CV cycles to +1.8V vs. RHE in 0.1M KOH. In the present manuscript both catalysts showed severe deactivation. The authors should comment on this apparent discrepancy with previous literature.

Firstly, our observation is consistent with the Reviewer's statement that CoFe₂O₄ is more stable than Co₂FeO₄. We observed that the activity of Co₂FeO₄ drops more rapidly compared to that of CoFe₂O₄; this is evidenced by the fact that the Tafel slope of Co₂FeO₄ rises from 43 ± 1 mV/dec in the pristine state to 83 ± 2 mV/dec after 1000 cycles, while the Tafel slope of

CoFe₂O₄ increases slightly from 79 ± 2 mV/dec (pristine) to 83 ± 1 mV/dec (1000 cycles). Essentially, the activity of CoFe₂O₄ degrades slightly and slowly (as also evidenced by our TEM and APT data that structural and compositional changes were observed for CoFe₂O₄ after 1000 cycles). We have mentioned this point in page 4: ‘Pristine Co₂FeO₄ has a Tafel slope of 43 ± 1 mV/dec (Fig. 1e), while pristine CoFe₂O₄ has a much larger Tafel slope of 79 ± 2 mV/dec (Fig. 1f)’ and ‘the Tafel slope of Co₂FeO₄ increases to 83 ± 2 mV/dec after 1000 cycles, which is almost double the Tafel slope of the pristine state, while the Tafel slope of CoFe₂O₄ increases slightly to 83 ± 1 mV/dec. Thus, despite the high OER activity of pristine Co₂FeO₄, its OER activity drops as the number of CV cycles increases, eventually reaching similar values as detected for the less active CoFe₂O₄.’

In comparison to our study, the previous work (J. Mater. Chem. A, 2018, 6, 7034) reported a higher activity loss for CoFe₂O₄ than for Co₂FeO₄. The discrepancy possibly arises from the difference in the synthesis procedure. In the previous study, the CoFe₂O₄ and Co₂FeO₄ nanoparticles were calcined at 450 °C and 900 °C, respectively, while we dried the nanoparticles at 180 °C for 12 h. Different synthesis procedures might induce a large discrepancy in the structure/chemistry of the pristine nanoparticles. This cannot be precisely evaluated because nanoscale characterisation, e.g. by APT, was not performed in the previous study. However, our speculation is indirectly supported by the evidence that the pristine CoFe₂O₄ and Co₂FeO₄ show similar current density (Fig. 5 in J. Mater. Chem. A, 2018, 6, 7034), while the current density of Co₂FeO₄ in our study is about an order of magnitude higher than that of CoFe₂O₄ (Fig. 1c-d in the main text). Therefore, a significant discrepancy of stability and activity of CoFe₂O₄ and Co₂FeO₄ during OER between the two studies is expected.

- (3) The main text and experimental section state that 1.0M KOH was used as an electrolyte, but the Figure 1 legend states 0.1M KOH. The authors should confirm the electrolyte concentration.

We apologise for the confusion. 1.0 M KOH was used instead of 0.1M. We have amended the caption of Fig. 1.

- (4) Supp. Figure 4 lists the BET-derived specific surface of the catalysts with a unit of m/g². Should this be m²/g?

We apologise for the typo. It has been amended.

- (5) The catalyst current density is normalised to BET-derived surface area. The authors should also report the geometric current density normalised to the electrodes geometric area.

We have added a new Supplementary Fig. 5 (see below) to show the linear sweep voltammetry and cyclic voltammetry data normalised to the geometric surface area of glassy carbon electrodes. The new figure shows a similar trend, as observed in Fig. 1a-d (in the main text). The overpotential of pristine Co₂FeO₄ (387 mV at 10 mA/cm²) is lower than that of CoFe₂O₄ (516 mV at 10 mA/cm²).

Supplementary Figure 5. (a, b) Linear sweep voltammetry and (c, d) CVs of CoFe_2O_4 and Co_2FeO_4 shown in Figs. 1a-d with current density normalised to the geometric surface area of the glassy carbon electrodes.

- (6) Page 4, line 120: authors report a Tafel slope of “ $\sim 83 \pm 1.7$ mV/dec” for Co_2FeO_4 after 1000 OER cycles. Why does this value have an approximate (\sim) designation since it has a reported standard deviation?

We thank the Reviewer for reading our work carefully. We have deleted ‘ \sim ’ in page 4. Also, we have removed digits after the decimal and updated Figs. 1e-f accordingly.

Reviewer #3 (Remarks to the Author):

This paper reports microstructure investigation of Co-Fe spinel oxides using APT method: it is possibly suitable for publication in this journal, but the authors should correct their lack of clarity.

The manuscript shows abundant observations including X-ray spectra and TEM images.

It also demonstrates multiscale investigations, combining chemical states at surface region by XPS, local co-ordinations by XAFS, phase separations by APT and etc.

The authors should mention the observing scales and regions of each technique briefly in the introduction or method sections. Such informations might help readers to understand the roles of techniques and the relations of their results.

We thank the Reviewer's constructive suggestion. We addressed this point by inserting text in the Introduction in page 3, and XPS and XAS result section in pages 6 and 7:

'Comprehensive information regarding the surface state changes is obtained by the scale-bridging method, including oxidation state measurements of bulk volume and top surface layer (5-10 nm) of nanoparticles by XAS and XPS, respectively, along with nanoscale and atomic-scale elemental and structural characterisation of individual nanoparticles by APT and HRTEM.'

'XPS measures the average oxidation state of approx. 100 μm x 100 μm x 5 nm of the surface region of the nanoparticles deposited on glassy carbon.'

'To further verify the irreversible change in the oxidation state of Co_2FeO_4 in their pristine state and after 1000 cycles, we performed XAS that allows spectral detection of a bulk volume of approx. 100 μm x 300 μm x 1 mm (penetration depth) of nanoparticles deposited on glassy carbon (Supplementary Note 1 and Supplementary Fig. 7).'

REVIEWERS' COMMENTS

Reviewer #1 (Remarks to the Author):

I thank the authors for providing a through response to sufficiently address the questions and concerns that were shared. I have no other outstanding issues.

Reviewer #2 (Remarks to the Author):

The authors have addressed my initial concerns and I believe the manuscript is now suitable for publication.